# Evidence of MHC class I and II influencing viral and helminth infection via the microbiome in a non-human primate

B. Karina Montero[1]*, Wasimuddin[2,3], Nina Schwensow[2], Mark A. F. Gillingham[2], Yedidya R. Ratovonamana[1,4], S. Jacques Rakotondranary[1,4], Victor Corman[5], Christian Drosten[5], Jörg U. Ganzhorn[1], Simone Sommer[2]

1 Institute of Zoology, Animal Ecology and Conservation, Universität Hamburg, Hamburg, Germany, 2 Institute of Evolutionary Ecology and Conservation Genomics, University of Ulm, Ulm, Germany, 3 Institute for Infectious Diseases, Faculty of Medicine, University of Bern, Switzerland, 4 Département Biologie Animale, Faculté des Sciences, Université d' Antananarivo, Antananarivo, Madagascar, 5 Institute of Virology, Charité-Universitätsmedizin Berlin, Berlin, Germany

* b.karina.montero@gmail.com

**Data Availability Statement:** The authors confirm that all data underlying the findings are fully available without restriction. MHC class I and class

## Abstract

Until recently, the study of major histocompability complex (MHC) mediated immunity has focused on the direct link between MHC diversity and susceptibility to parasite infection. However, MHC genes can also influence host health indirectly through the sculpting of the bacterial community that in turn shape immune responses. We investigated the links between MHC class I and II gene diversity gut microbiome diversity and micro- (adenovirus, AdV) and macro- (helminth) parasite infection probabilities in a wild population of non-human primates, mouse lemurs of Madagascar. This setup encompasses a plethora of underlying interactions between parasites, microbes and adaptive immunity in natural populations. Both MHC classes explained shifts in microbiome composition and the effect was driven by a few select microbial taxa. Among them were three taxa (*Odoribacter*, *Campylobacter* and Prevotellaceae-UCG-001) which were in turn linked to AdV and helminth infection status, correlative evidence of the indirect effect of the MHC via the microbiome. Our study provides support for the coupled role of MHC diversity and microbial flora as contributing factors of parasite infection.

## Author summary

The selective pressure of the major histocompatibility complex (MHC) on microbial communities, and the potential role of this interaction in driving parasite resistance has been largely neglected. Using a natural population of the primate *Microcebus griseorufus*, we provide correlative evidence of two outstanding findings: that MHCI and MHCII diversity shapes the composition of the gut microbiota; and that select taxa associated with MHC diversity predicted adenovirus and helminth infection status. Our study highlights the importance of incorporating the microbiome when investigating parasite-mediated MHC selection.

II gene sequences were deposited in GenBank under accession codes MT776952-MT777176 and MT776908-MT776951, respectively. The Sequence Read Archive (SRA) accession ID of the 16S rRNA gene sequences reported in this paper is SRP217185. Data supporting the findings of this study and R scripts have been deposited with figshare under https://doi.org/10.6084/m9.figshare.12659927.

**Funding:** JUG was funded by the Bundesministerium für Bildung und Forschung SuLaMa/BMBF (FKZ 01LL0914) (https://www.bmbf.de/en/research-funding-1411.html) JUG was funded by the Deutsche Forschungsgemeinschaft, DFG, SPP 1596 "Ecology and Species Barriers in Emerging Viral Diseases", Ga 342/19-1, (https://www.dfg.de/en/research_funding/index.html) The funders had no role in study design, data collection and analysis, decision to publish, or preparation of the manuscript.

**Competing interests:** The authors have declared that no competing interests exist.

# Introduction

Reciprocal interactions between the hosts' immune system and the gut microbial flora are essential in defining the course of a parasitic challenge for three main reasons. First, specific bacterial taxa can prime immune signaling that provoke the activation of an inflammatory response against parasites [1, 2]. Second, the secretion of inhibitory substances (e.g. antimicrobials) and metabolites by some bacteria have been shown to inhibit growth and the ability of pathogens to attach to the intestinal lumina [3, 4]. Third, the competitive advantage of commensal bacterial taxa over a niche can limit the growth and expansion of pathogenic bacteria [5, 6]. Evidence of these and other forms of colonization resistance underscore the relevance of immune mechanisms that encourage a balanced state between tolerance towards commensal bacteria and resistance against offending invaders [7, 8]. An important factor influencing this balance is the selective sculpting of bacterial communities by means of adaptive immunity [9–11]. The involvement of the major histocompatibility complex (MHC) in antigen-specific immune response appears to contribute to microbial colonization [12–14], however the interplay between the natural variation of the MHC and microbial communities in parasite resistance has yet to be explored.

The MHC is a gene-dense region with loci coding for molecules involved in antigen-specific responses (such as the classical MHC genes: MHC class I and MHC class II) as well as for proteins that do not participate in antigen processing but have other immune and cell signaling functions (including MHC class III genes, such as complement components factor B, C2 and C4, the cytokines Tumor necrosis factor and lymphotoxins, and nonclassical MHC genes) [15, 16]. Whereas the function of many MHC nonclassical and class III genes remain poorly characterised, particularly in non-model organisms [17, 18], classical MHC class I and II genes have been of focal interest in the study of adaptive immunity and susceptibility to infection in many taxa due to their known antigen-presenting function [19, 20]. Both MHC class I and class II molecules are part of two distinct antigen processing pathways contingent on the nature of the foreign peptide. MHC class I molecules process intracellular peptides (e.g. viruses and cancer cells) and present them to CD8[+] T-cells while extracellular peptides (e.g. bacteria and helminths) are presented to CD4[+] T-cells by MHC class II molecules [15, 21]. The high affinity of CD4[+] T-cells to the bacterial antigen/MHC class II complex trigger inflammatory [1, 22, 23] and tolerogenic [24] responses, which in turn keep commensals in check. Therefore, most studies focus on the role that the MHC class II pathway plays in the gut microbiome. Evidence of compositional differences in the microbiome associated to primed CD8[+] T-cells by MHC class I molecules is scarce, although the MHC class I pathway synergizes with specific bacterial taxa to mediate parasite resistance [2, 25] and tumor development [26]. Furthermore, processing of extracellular antigens through the specialized role of dendritic cells for cross-presentation [27] has been associated with potent T-cell responses [28]. Therefore, both CD4[+] and CD8[+] T-cells are likely involved in tolerogenic or immune responses towards parasites and gut commensals.

Here, we used a natural population of the reddish-gray mouse lemur *Microcebus griseorufus* to reconcile the inter-dependent relationship of MHC class I, class II and microbial diversity on micro- and macro- parasite infection propensity (a detailed summary of the study design and hypotheses are given in Fig 1. To this end, we explored adenovirus (AdV) and helminth infection status. AdV is a DNA virus causing widespread infections and are considered a major concern for human health globally [29]. Although AdV infection is often asymptomatic, it can develop into severe respiratory and gastrointestinal diseases in children, immunosuppressed patients and in the elderly [30]. It is also important to monitor AdV prevalence in

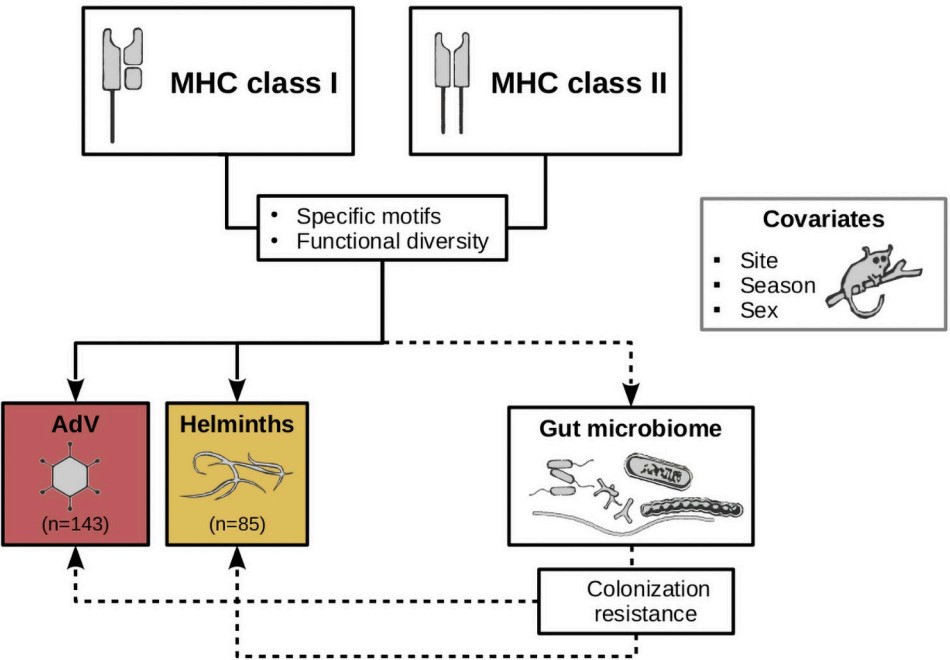

**Fig 1. Study design.** The role that MHC polymorphisms may play on parasite infection status in natural populations can be better understood under the lens of their direct and indirect contribution. The MHC may have direct effects on parasite infection via antigen-driven immune response. It may also indirectly affect the environment that parasites face when establishing an infection by shaping the gut microbiome. We evaluated the direct (solid arrows) and indirect (dashed arrows) contribution that MHC class I and class II diversity may have on micro- (adenovirus, AdV) and macro-parasite (helminths) infection status in a wild population of non-human primates, the mouse lemur *M. griseorufus*, while controlling for the effect of extrinsic (sampling site and season) and intrinsic (sex) factors. We assessed MHC diversity in terms of the presence of specific motifs (MHC class I supertypes and MHC class II alleles) and functional diversity (sequence divergence and number of supertypes). We evaluated the direct effects of the MHC on infection status according to the following predictions: i) specific MHC class I supertypes and MHC class II alleles will be associated with susceptibility (or resistance) to AdV and helminth infection, respectively; ii) functional diversity will be negatively associated with parasite infection status. For the indirect effects of the MHC on infection status via the microbiome, we predicted that MHC I and MHC II diversity influences the diversity and the composition of the microbiome. In particular we predicted that specific MHC motifs and MHC functional diversity will be negatively associated with the relative abundance of pathogenic taxa. In contrast, we expect to find that taxa not subject to immune recognition mediated by the MHC are involved in essential metabolic functions and contribute to resistance to parasitic challenges.

wildlife, in particular in non-human primates, since interspecies transmission are known to occur and can impact human health [31, 32]. In contrast, helminths play important immuno-modulatory roles in the establishment of long-lasting infections that can be either harmful, particularly for undernourished and immunosuppressed individuals [5], or beneficial to the host by limiting the development of immune pathology, a process that involves the concerted action with the microbiota [33, 34]. Viral infection, helminths and commensal bacteria share the common agenda of promoting acceptance by its host, often via immune regulation. Our group previously demonstrated a link between AdV and gut microbiome using 16SrRNA gene amplicon data [35]. Given its central role in immunity, we expand on our previous findings and tested the hypothesis that diversity at the MHC is an important component of the host's genetic landscape in shaping gut microbial communities, which in turn influence micro- and macro- parasite infection.

## Results

### The microbiome predicts infection status

Consistent with previous work on mouse lemurs [32, 36], AdV and helminth prevalence in our study population was high; 29.4% of individuals tested $AdV^{pos}$ (42 from the 143 individuals screened for AdV infection), 40.1% of individuals (35 from the 85 individuals screened for helminth infection) were infected with helminths, and 8.6% of individuals were co-infected with AdV and helminths (10 from the 85 individuals screened for both AdV and helminth infection). We found no statistical support for AdV infection to predict helminth infection and *vice versa* ($\chi^2$ = 0.841; p-value = 0.35). We detected a total of 226 MHCI functional alleles (median of 7 alleles per individual, range 4–20) in the population and based on physicochemical variables (z-values [37]) of the amino acid sites affected by positive selection (PSS) and grouped them into 17 MHC supertypes (ST) (Materials and methods, S1 Fig and S1 Table). For the MHCII, we identified 49 functional alleles. A large number of MHCII alleles were rare (present in 1–2 individuals), therefore we used for subsequent analyses only alleles that were found in more than three individuals (S1 Fig). We found that neither MHCI nor MHCII motifs were associated with AdV nor helminth infection status (p-values from odds ratio tests after Bonferroni multiple testing adjustment > 0.05; S3C–S3F Fig).

We used binomial generalized linear models (GLMs) to examine the effect of MHC and microbiome diversity on parasite infection while controlling for the potential effects of covariates (site, season and sex). We performed model selection using the information theoretic (IT) approach and we estimated the effect of predictors using weighted model averaging (see Materials and methods) [38]. We used the number of MHCI supertypes ($MHCI_{nST}$), mean amino acid allele divergence over PSS at MHCI gene ($MHCI_{distPSS}$), and mean amino acid allele divergence over PSS at MHCII gene ($MHCII_{distPSS}$) as proxies of MHC functional diversity (Materials and methods, S4 Fig). For the microbiome we distinguished between diversity (Faith's PD and Shannon's index of diversity) and divergence (Bray-Curtis distance from the population mean). The former quantifies gut microbial diversity within an individual, whilst the latter quantifies how atypical a host's microbiome composition is relative to the population mean.

We found no support of an association between MHC functional diversity and AdV or helminth infection status according to model selection (S2–S5 Tables). However, our study provides evidence of an association between microbiome diversity and divergence with AdV and helminth infection, respectively. AdV infection was associated with an increase in Faith's PD observed at both study sites (Δ AIC = 6.76; partial-r [±95%CI] = 0.23 [0.12, 0.34]) (Fig 2A, S2 and S4 Tables). We found a negative relationship between microbiome divergence and helminth infection status (Δ AIC = 2.26; partial-r [±95%CI] = -0.22 [-0.41, -0.04]) (Fig 2D, S2 and S4 Tables), indicating that gut microbial communities across helminth infected individuals are more similar than the communities of non-infected individuals. A relationship between Shannon's diversity index and AdV or helminth infection probability was not supported by model selection (S3 and S5 Tables). With respect to the influence of other covariates on parasite infection, we found support for the effect of site on AdV infection status (Δ AIC = 3.05; Cohen's D [±95%CI] = -0.42 [-0.74, -0.09]) (Fig 2B, S2–S5 Tables); individuals caught in the sampling site Andranovao were more likely to be $AdV^{pos}$ than in Miarintsoa. In addition, we found support for an effect of season on helminth infection status (Δ AIC = 5.19; Cohen's D [±95%CI] = -0.67 [-1.18, -0.15]) (Fig 2D, S2–S5 Tables); the probability of infection was higher among individuals captured during the dry season, as has been demonstrated in baboons [39]. Limited food availability during the dry season relative to the wet season results in nutritional stress which is considered an important driver of increased parasite burden [40].

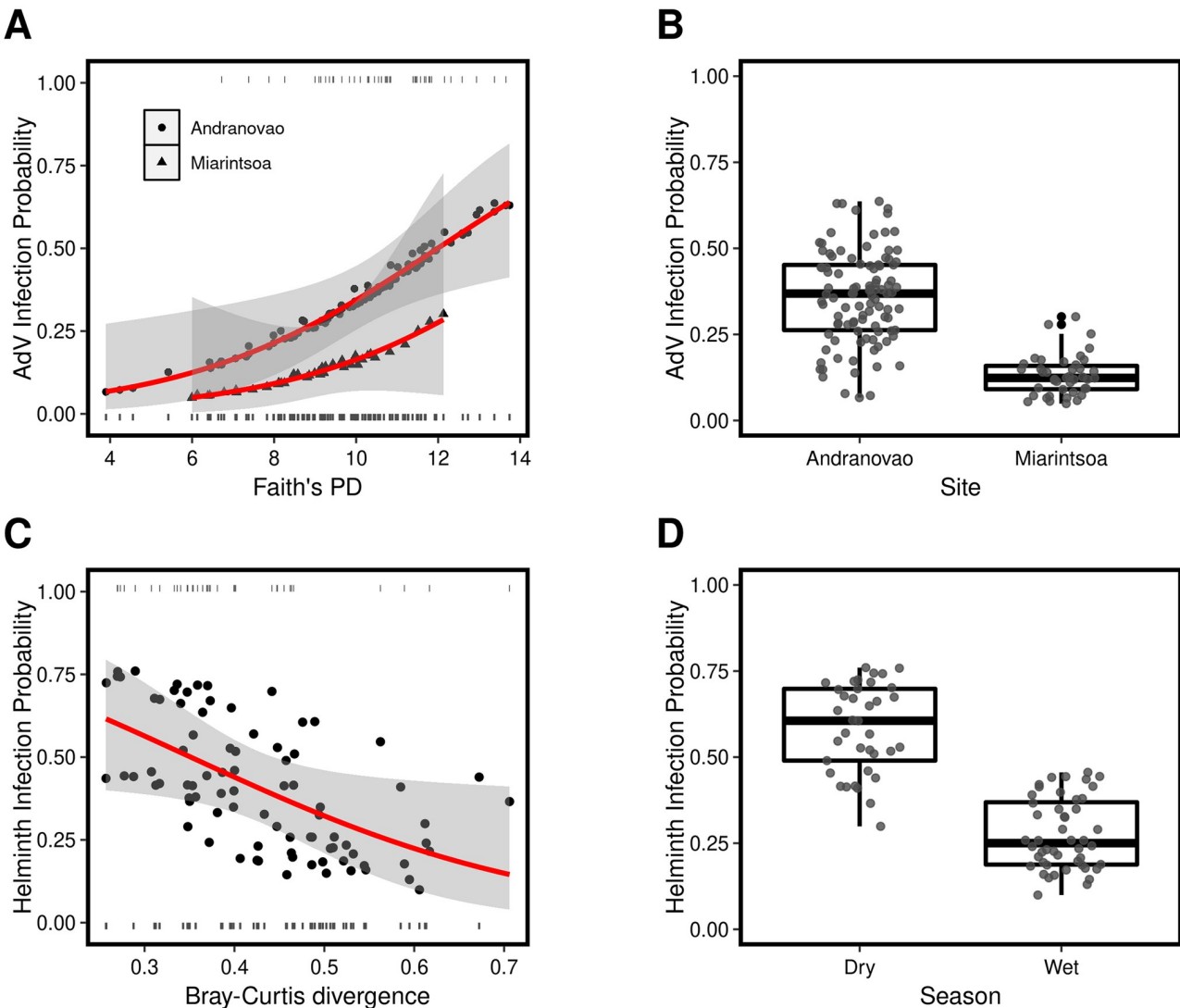

**Fig 2. Association between the gut microbiome, covariates, and AdV and helminth infection.** Faiths' PD and site were retained as predictors of AdV infection status by model selection, whilst Bray-Curtis divergence and season were retained as predictors of helminth infection status. AdV infection probability according to model averaged predicted values of *(A)* Faiths' PD and *(B)* sampling site. Helminth infection probability according to model averaged predicted values of *(C)* Bray-Curtis divergence and *(D)* season. Fitted lines in *(A)* and *(C)* are shown in red and 95% confidence intervals are shaded in gray. The rug represents individuals that were infected (a value of 1) or not infected (a value of 0), whilst the filled symbols represent the predicted values from the GLMs.

## MHC diversity is associated with a shift in the composition of the gut microbial community

To investigate the indirect role of the MHC on parasite infection via the microbiome, we explored the association between MHC diversity and microbiome diversity and composition. Using regression models we found no evidence that MHC diversity is associated with microbial diversity within an individual (S6 Table). In contrast, gut microbiome composition was associated with MHC diversity. We used canonical correspondence analysis (CCA) analysis to explore the contribution of MHC functional divergence, specific MHC motifs and covariates shaping the composition of the gut microbial community. Our results revealed that sequence

divergence at $MHCI_{distPSS}$ ($r^2 = 0.16$, $P<0.0001$) (but not $MHCI_{nST}$) was a significant predictor of distance between samples in the constrained ordination space (S5 Fig). Contrary to expectations based on the affinity of class I vs class II to extracellular peptides, we did not find evidence of an effect of MHCII diversity on microbial composition ($r^2 = 0.03$, $P = 0.09$). We observed shifts in microbial composition according to the presence of eight MHCI supertypes (S6 Fig) and four MHCII alleles (S7 Fig). Regarding the effect of parasite infection on composition, both AdV infection status ($r^2 = 0.03$, $P = 0.01$; S8A Fig) and helminth infection predicted a shift in microbial composition ($r^2 = 0.05$, $P = 0.02$; S8B Fig). The former result replicates our previous findings that AdV[pos] individuals have a different microbiome from AdV[neg] individuals [35]. Ordination analyses also reveal a significant effect of season ($r^2 = 0.37$, $P<0.0005$; S8C Fig) and sampling site ($r^2 = 0.21$, $P<0.0005$; S8D Fig), highlighting the fine-scale temporal and spatial patterns of microbial diversity [41]. Sex had no effect on beta diversity ($r^2 = 0.005$, $P = 0.47$). Our study sites differ sharply in vegetation mostly as a consequence of differences in precipitation regimes. The strong effects of site and seasonal on gut microbiome composition are likely to reflect differences in food availability and dietary preferences [42, 43].

We identified core taxa (prevalence > 60%, i.e. the ASV was present in > 60% of individuals regardless of abundance) associated with specific MHCI supertypes and MHCII alleles (using analysis of composition of microbes (ANCOM)) and that differed most with respect to their association with high or low estimates of MHC functional diversity (using a differential ranking approach) (see Materials and methods). We found that both MHCI and MHCII diversity were linked to shifts in the relative abundance of a limited number of core taxa (Fig 3). In fact, except for one MHCII allele (Migr-DRB*36) which was associated with seven core amplicon sequence variants (ASVs) (and with 22 ASVs in total) and was strongly associated with an overall shift in microbial composition ($r^2 = 0.16$, $P<0.001$; S7C Fig), the vast majority of MHCI supertypes and MHCII alleles were associated with a small range (1–3) of ASVs (Fig 3A and 3B). Two possible hypotheses may explain how a particular allele interacts with the microbiome. Either Migr-DRB*36 is a generalist allele [44] that indiscriminately binds a high number of antigens affecting both commensal and potentially pathogenic taxa; or Migr-DRB*36 is a specialist that binds to antigens of a core taxon that plays a central role in the microbial community, leading to a shift in the relative abundance of other taxa. Indeed 5 out 7 of the core taxa had lower relative abundance among individuals that carried Migr-DRB*36.

## Evidence of an indirect effect of the MHC on parasite resistance

We then used log-ratios, a suitable method for examining differences in abundance using compositional data [45], to estimate differences in relative abundance of the ASV's associated to MHC diversity (MHC motifs and functional diversity) according to AdV and helminth infection status. We used *Bifidobacterium* as a reference frame since this taxon was present in most samples, and is expected to be a stable member of the microbial community (Materials and methods). Whilst causality cannot be inferred from correlation, our analyses nonetheless suggest that resistance to AdV was provided by an increase in *Campylobacter* (Fig 4B) which was associated with both MHCI and MHCII diversity (MHCI supertypes: ST*3; MHCII allele Migr-DRB*36 and $MHCI_{nST}$, Fig 3A, 3B and 3C), an increase in *Olsenella* (Fig 4C), a microbe thought to prevent inflammation [46] associated to $MHCI_{nST}$ (Fig 3C), and an increase in abundance of *Odoribacter* (Fig 4D), a microbe enriched among individuals with the MHCI supertype ST*2 (Fig 3A). Furthermore, the ASV assigned to the Prevotellaceae UCG-001 group and that was associated to several MHC motifs (MHCI supertypes: ST*1 and ST*8; MHCII alleles: Migr-DRB*36 and Migr-DRB*29, Fig 3A and 3B) was linked to helminth infection (Fig 4E).

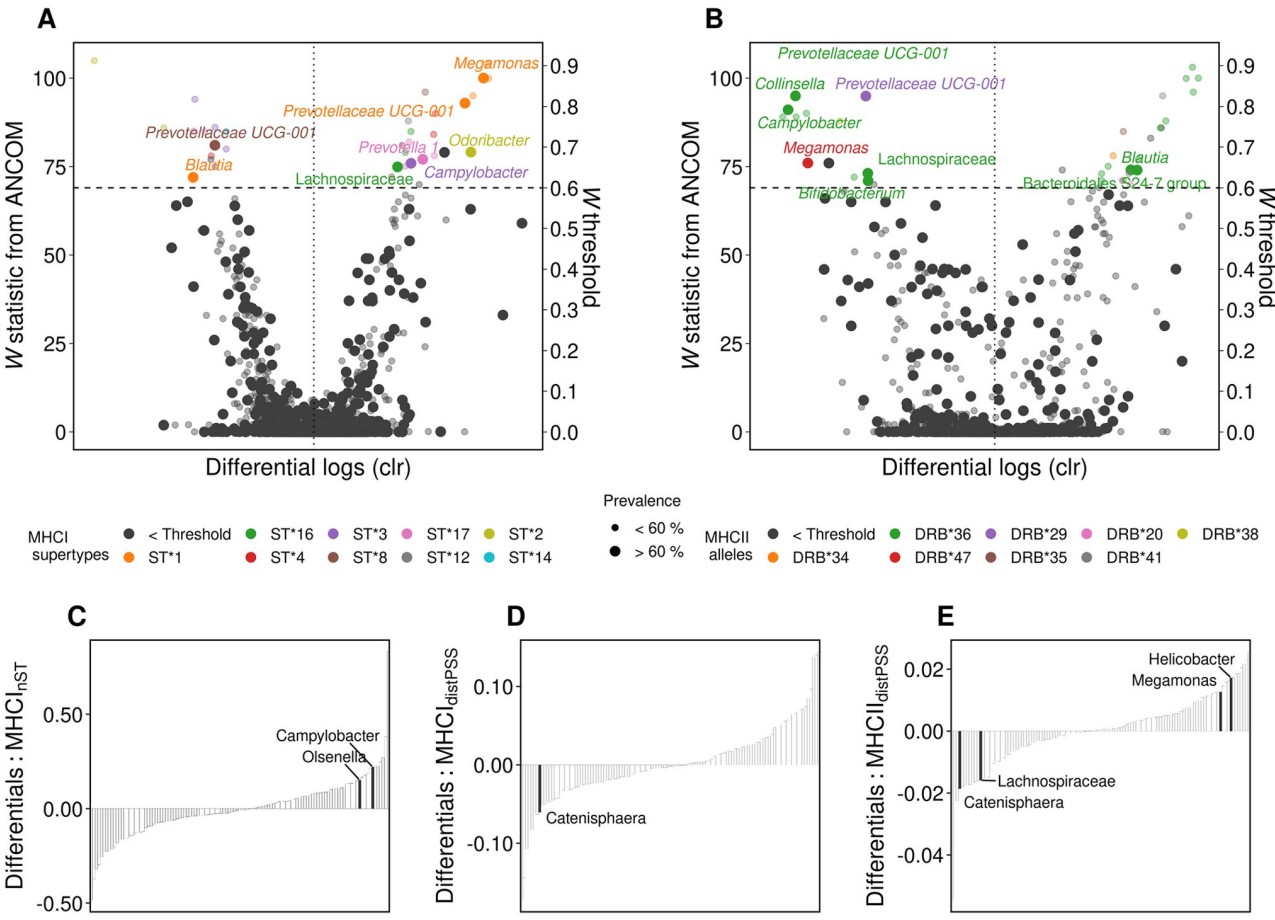

**Fig 3. MHC diversity is associated with shifts in microbiome composition.** *(A-B)* Association between specific MHCI supertypes and MHCII alleles and microbial taxa (n = 143). Significance threshold is informed by *W* statistic (horizontal dashed line) and *F* estimates (vertical dotted line). ASV's above this threshold represent taxa significantly ($P<0.005$) associated with a specific MHC motif. Taxa are colored according to their association with a MHC motif. Labels are shown for core taxa (prevalence >60%). *(C-E)* Microbial ranks based on multinomial regression coefficients sorted by their association with supertype diversity (MHCI$_{nST}$), MHCI sequence divergence (MHCI$_{distPSS}$) and MHCII sequence divergence (MHCII$_{distPSS}$) (n = 143). Labels correspond to high and low rankings of core taxa.

Overall, more taxa associated with MHC diversity were also linked to AdV resistance compared to the number of taxa influencing helminth infection, suggesting a stronger indirect effect of the MHC on AdV infection propensity than helminth infection status. Notably, among the taxa associated with infection, both *Campylobacter* and *Odoribacter* had a particularly strong effect on AdV propensity (Fig 4A). *Campylobacter* is a genus well known for its high number of important pathogens for human and animal health [47], whilst both *Odoribacter* and Prevotellaceae UCG-001 are known to generate short-chain fatty acids (SCFA) [48, 49].

## Links between MHC diversity and diversity of functional bacterial pathways

To evaluate if the observed shift in taxonomic composition of the microbiome influences functional attributes, we generated and analyzed the abundance of MetaCyc pathway predictions using the PICRUSt2 algorithm [50]. MetaCyc is an open-source database for metabolic

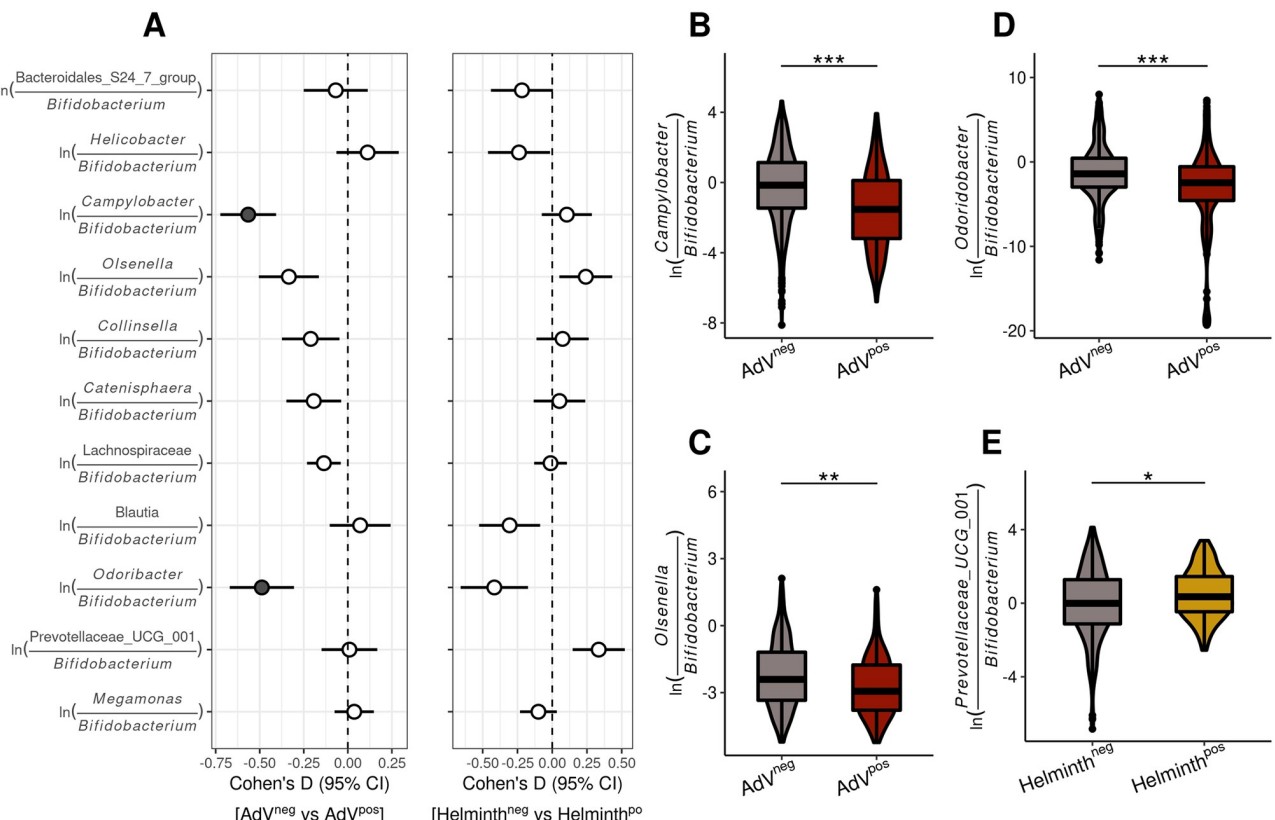

**Fig 4. Bacterial taxa associated with MHC motifs and functional diversity are significantly associated with AdV and helminth infection status.** *(A)* Effect size (Cohen's D and CI) forest plots of the log-ratios of taxa associated to MHC motifs and to MHC functional diversity with respect AdV and helminth infection. Filled symbols represent log- ratios with medium to large effect sizes (Cohen's D |>0.50|). ASVs of the genus *Bifidobacterium* were used as a reference frame to calculate the log-ratios. *(B-E)* Violin box-plots illustrating the significant log-ratio across AdV (n = 143) and helminth (n = 85) infection status. * $P < 0.05$; ** $P < 0.005$; *** $P < 0.0005$.

pathways based on organismal annotated genomes [51]. We first evaluated whether MHC functional diversity and covariates predicts pathway diversity (pathway richness and evenness using Shannon's index) and composition. We included a total of 296 MetCyc pathways in the analysis. Greater peptide recognition among individuals with high levels of MHC functional diversity can constrain the ability of a variety of bacterial taxa to colonize the gut, a scenario that would be supported by a negative linear coefficient in functional microbial diversity. In contrast, a positive linear coefficient would suggest that maximal MHC diversity is associated with the maintenance of high functional microbial diversity as a result of the removal of spe-cific taxa that negatively impact other members of the microbial community. We found that MHCII sequence divergence was negatively associated with microbiome pathway evenness (Δ AIC = 2.26; partial-r [±95%CI] = -0.22 [-0.41, -0.04]) (Fig 5C), S7 Table), suggesting that MHCII functional divergence limits gut microbial functional diversity. The fact that MHCII sequence divergence was associated with pathway evenness (Shannon's diversity Index) but not pathway richness, suggests that the effect of the MHC was stronger on abundant core path-ways than on sparse pathways. None of the MHCI estimates were associated with pathway diversity (S7 Table), suggesting that MHCI molecules do not influence the overall functional diversity of the microbial community. In contrast to taxonomic based analyses, AdV and hel-minth infection status had no effect on pathway diversity (S7 and S8 Tables). With respect the

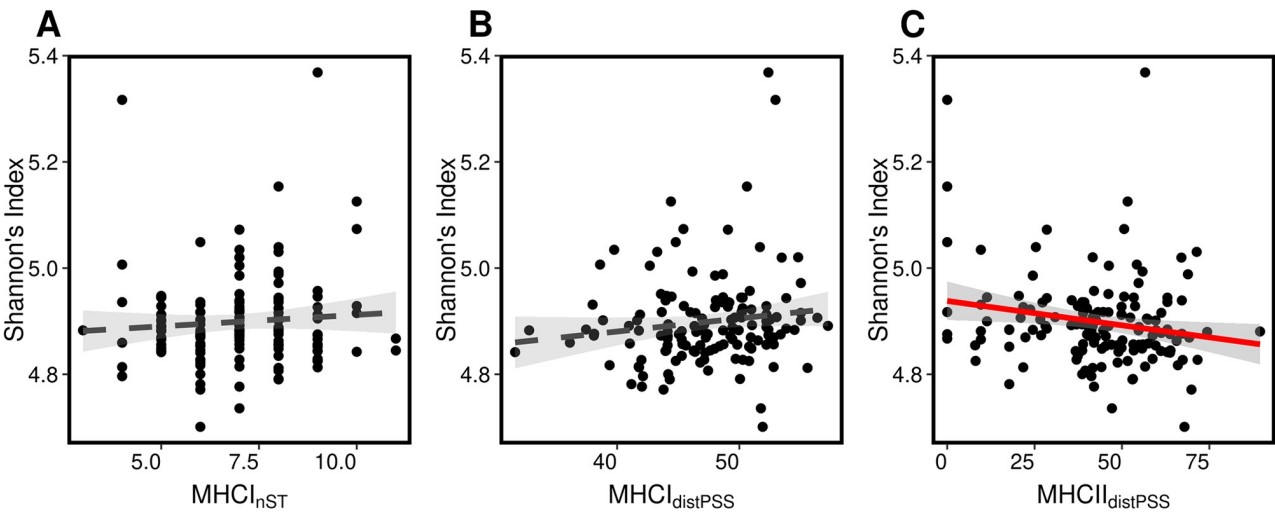

**Fig 5. MHC is associated with functional diversity of the microbiome.** Relationship between *(A)* number of MHCI supertypes, *(B)* MHCI sequence divergence and *(C)* MHCII sequence divergence and the functional diversity of the microbiome according to Shannon's diversity index. Associations supported by model selection ($\Delta$ AIC > 2) are reported with a red solid line, shaded area represent 95% confidence intervals. Dashed lines represent non-significant trends.

effect of covariates, we found support for temporal variation in pathway eveness; pathway eveness in the second dry season of sampling was higher than in all other sampled seasons ($\Delta$ AIC = 15.9; Cohen's D $_{Dry2\ vs\ Dry1}$ [±95%CI] = -0.70 [-1.08, -0.31], Cohen's D $_{Dry2\ vs\ Wet1}$ [±95%CI] = -0.77 [-1.17, -0.38], Cohen's D $_{Dry2\ vs\ Wet2}$ [±95%CI] = -0.51 [-0.88, -0.13], S9A Fig). In addition, we found an effect of sampling site on pathway richness ($\Delta$ AIC = 7.02; Cohen's D [±95%CI] = -0.58 [-1.00, -0.14], S9B Fig), further emphasising previously mentioned habitat specific effects on the gut microbiome.

CCA analysis indicate that the MHC had limited effects on the overall composition of metabolic pathways. On the one hand, we found fewer associations between specific MHC motifs (2 MHCI supertypes and 1 MHC II allele, S10 Fig) and among-individual distance in the ordination space compared to taxonomic-based analyses. On the other hand, none of the MHC metrics of functional diversity were significant predictors of the composition of pathways. Furthermore, consistent with taxonomic based analyses we also detected significant temporal variation ($r^2$ = 0.15, $P$<0.0005) in the composition of functional pathways (S9C Fig).

As a next step we identified pathways whose relative abundance was associated to specific MHC motifs and pathways with high and low ranks according to MHC functional diversity and then used log-ratios to test for differences among AdV and helminth infected vs non-infected individuals. To this end, we used ANCOM and differential ranking as described above for taxonomic-based analyses. We found that a particular set of MHC motifs were linked to differences in the relative abundance of a restricted number of pathways (Fig 6A and 6B). Mirroring our results on bacterial taxa, most motifs were associated with a few pathways (1–9), with the exception of ST*3 and Migr-DRB*36 which were associated with differences in the relative abundance of a larger number of pathways (14 and 12 pathways, respectively). Finally, multinomial analyses revealed that low MHCI and MHCII functional diversity (for both number of supertypes and sequence divergence) was associated with relatively rare pathways, while core pathways were associated with high MHCI and MHCII sequence divergence (Fig 6C–6E).

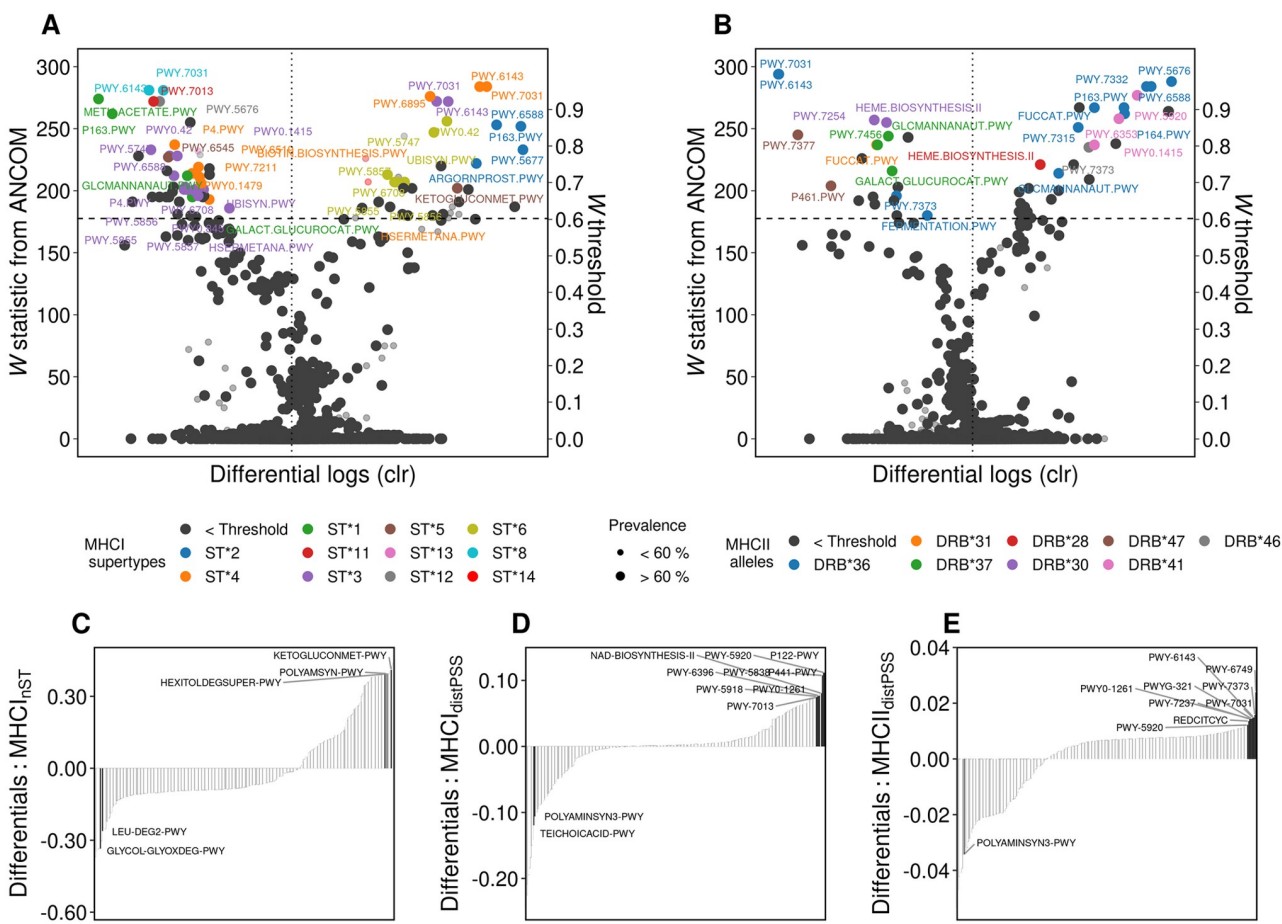

**Fig 6. MHC diversity is associated with composition of functional pathways.** Association between specific MHCI supertypes and MHCII alleles and metabolic pathways *(A-B)*. Significance threshold is informed by W statistic (horizontal dashed line) and F estimates (vertical dotted line). Metabolic pathways above this threshold represent pathways significantly ($P < 0.005$) associated with a specific MHC motif. Pathways are colored according to their association with a MHC motif. *(C-E)* Microbial pathway ranks based on multinomial regression coefficients sorted by their association with supertype diversity ($MHCI_{nST}$), MHCI sequence divergence ($MHCI_{distPSS}$) and MHCII sequence divergence ($MHCII_{distPSS}$). Labels correspond to high and low rankings of core pathways (>60% prevalence).

To calculate log-ratios we used the TCA pathway (PWY-7254) as a reference since this is a common pathway central to the generation of energy and present in most aerobic living organisms. We did not find significant differences in the relative abundance of pathways according to parasite infection. However for this analysis we had large effect size confidence intervals (as a result of strong inter-individual stochasticity in pathway abundance) suggesting a lack of statistical power. For instance, when considering pathways with medium to large effect sizes (| Cohen's D| > 0.5), we identified three pathways with a lower relative abundance among individuals testing positive for AdV and helminth infection (Fig 7A). The relative abundance of pathways involved in the generation of glycan (PWY-7031) and sugar (PTW-6143) were increased among AdV^neg individuals (Fig 7A), suggesting that these pathways are involved in limiting AdV infection. Interestingly, the stratified pathways revealed that *Campylobacter* contributes entirely to the diversity of these pathways (Fig 7B and 7C). Indeed, PWY-7031 and PTW-6143 have been associated with adherence and pathogenicity, which suggests that *Campylobacter*-enriched functional pathways associated with high MHCII sequence divergence and multiple MHC motifs (MHCI supertypes: ST*1, ST*4 and S*8; MHCII allele Migr-

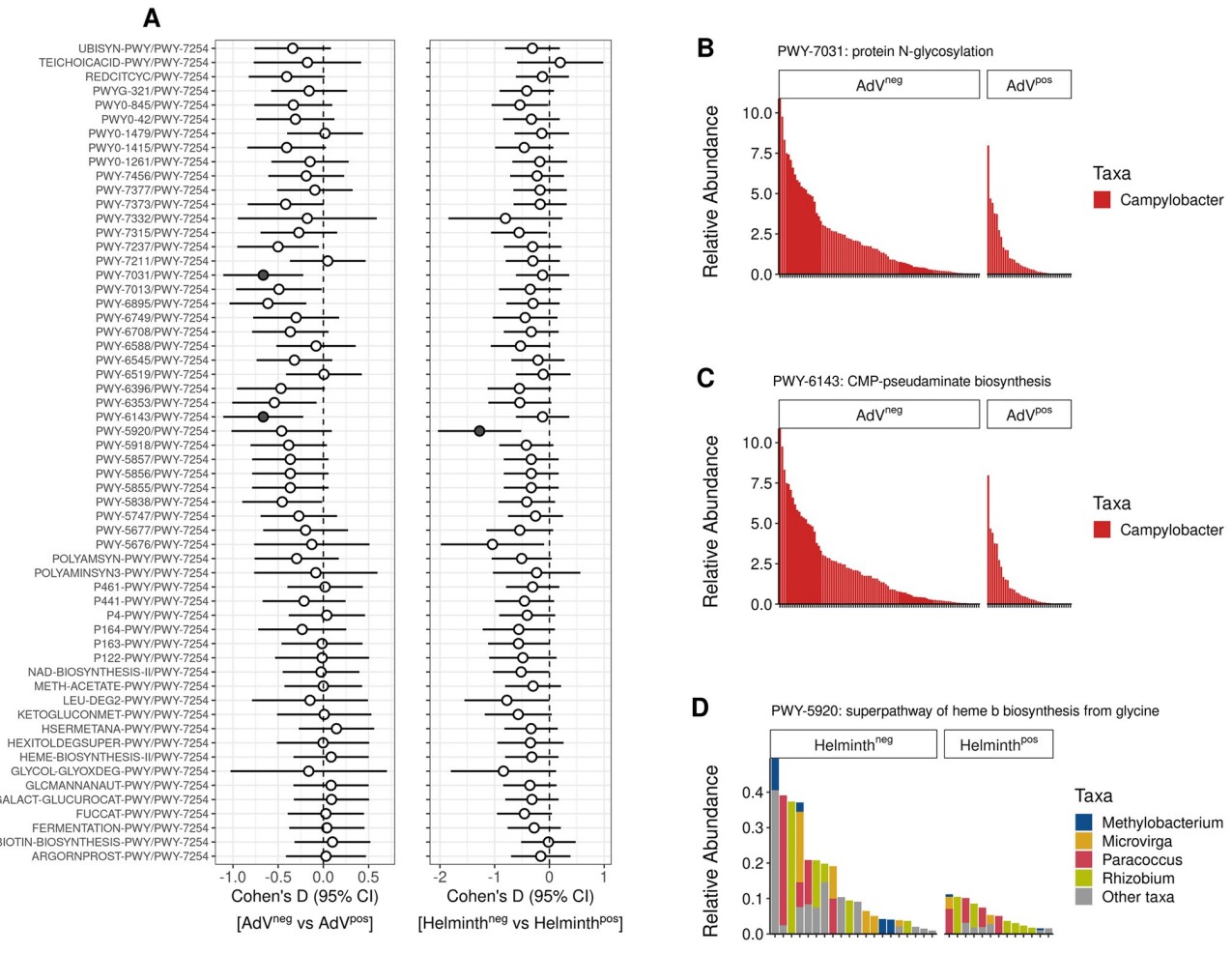

**Fig 7. Differences in relative abundance of functional pathways associated with MHC diversity according to AdV and helminth infection.** *(A)* Effect size (Cohen's D and CI) forest plots of the log-ratios of the functional pathways associated to MHC motifs and to MHC functional diversity with respect to AdV and helminth infection. The pathway PWY-7254 was used as a reference frame to calculate the log-ratios. Filled symbols represent log-ratios with medium to large effect sizes (|Cohen's D| > 0.50). *(B-D)* Contribution of bacterial taxa to pathways with medium-large effect sizes according to AdV or helminth infection. Stacked bar plots indicate the relative abundance of taxa across samples.

DRB*36) reduce AdV susceptibility. A single metabolic pathway (PWY-5920), decreased in abundance among helminth[neg] individuals (Fig 7D). This pathway is involved in heme biosynthesis, an iron prosthetic group that is essential in the development and reproduction of helminths [52].

## Discussion

The contribution of MHC diversity on parasite resistance has traditionally been evaluated as a direct interaction. Indeed, parasite mediated selection is considered the main mechanism driving the extraordinary polymorphism of MHC genes [20, 53]. However, parasites do not only face the challenge of avoiding immune recognition, but also interact with a multitude of host-associated microbes that influence their ability to successfully colonize the host [33, 54–56]. Our study identifies a link between MHCI and MHCII diversity and the gut microbiome of mouse lemurs, and in turn taxa associated with MHC diversity predicted parasite infection.

Our findings support the notion that natural diversity of the MHC is associated to parasite resistance via the microbiome.

Parasite-specific immune responses driven by MHC polymorphisms is extensively supported [57–60], however in our study we found no evidence of an association between MHC diversity and AdV or helminth infection status. The latter may be the consequence of a lack of statistical power [61] and/or the fact that we are analysing infection status without accounting for specific strains or infection intensity. Another possible caveat in association studies is the use of supertypes to cluster alleles based on functional properties, as done here with the MHCI of mouse lemurs. In systems with highly duplicated loci, allelic diversity is so high that it would require unfeasibly large sampling effort to detect associations between pathogens and a specific allele. The classification of alleles into supertypes can therefore circumvent this limitation if a supertype is represented by overlapping peptide binding repertoires, as has been shown for several human leukocyte antigen (HLA) class I molecules [62, 63]. However, closely-related alleles (as little as a difference in a single amino acid) can have highly contrasting outcomes in terms of disease susceptibility [64, 65]. Thus, since a single supertype can potentially harbour several alleles with opposing immunological effects, applying the supertype approach might reduce our resolution in detecting significant associations. In contrast to specific MHC motifs, we did not expect a strong association between disease susceptibility and MHC functional diversity. This is because MHC functional diversity is more likely to be associated with parasite diversity than infection status of a single parasite [66].

Concerning the relationship between infection status and the microbiome, we found a link between between AdV and helminth infection and distinct aspects of microbial diversity: AdV was associated with the diversity of the microbiome, while helminth infection was related to the divergence of the microbial community. As has been previously suggested [35], AdV can have a direct effect on the microbial community. The reciprocal interaction between the hosts' immune response, gut microbiome and parasites limits our ability to infer the direction of causality using the correlative approach of our study [67]. Therefore, our results suggest that gut microbiome diversity constitute an important element in explaining parasite infection status.

Given the immunological challenges stemming from high microbial diversity in the gut, we expected that MHC diversity would shape microbiome diversity and composition. We found a subtle negative effect of MHCII sequence divergence (i.e. MHCII functional diversity) on the diversity of metabolic pathways (i.e. predicted functional diversity of the microbiome) ($r^2 =$ 4.1%). However, we did not find evidence of an effect of MHC diversity on microbiome diversity at the taxonomic level. Each metabolic pathway is characterised by a set of bacterial taxa (resulting in functional redundancy [68]), suggesting that the effect of MHCII was only observed in pathway-based analyses due to the cumulative effect of combining multiple taxa. We hypothesize that there is a cost to broad antigen-derived immune responses through high MHCII diversity by negatively impacting functional microbial diversity. A cost has previously been suggested through the lens of autoimmune disease [69], whereby there is a trade-off between maximal parasite recognition and the risks associated with autoimmune diseases. Thymic negative selection eliminates T-cells with strong binding affinity to self/MHC complexes [70, 71]. Consequently, high MHC diversity may limit the T-cell receptor repertoire (TCR), leading to suggestions that an optimal intermediate MHC diversity should be selected rather than maximised. However, the trade-off between thymic selection and MHC diversity has been challenged since increased MHC diversity might also enhance positive selection of the TCR repertoire [72]. Thus to date, the mechanisms that constrain MHC diversity remain an enigma [20]. Our recent understanding of microbial diversity within the host requires MHC diversity theory to be revised to account for the potential cost of immunity on the metabolic activities of the microbiome and merits future empirical research.

We evaluated qualitative (specific MHC motifs) and quantitative (MHC functional diversity) MHC effects on gut microbiome composition. We found that both MHC genes were associated with shifts in the composition of the microbial community, with supertype MHCI ST*17 ($r^2$ = 9.6%) and MHCII allele Migr-DRB*36 ($r^2$ = 16.0%) having particularly strong effects. MHCI, but not MHCII, sequence divergence was also linked to among individual differences in microbial communities ($r^2$ = 16.4%). We also note that whilst we found an effect of MHCI sequence divergence, the number of MHCI supertypes was not a significant predictor of microbial composition. Although sequence divergence has been shown to be correlated with peptide binding repertoire, the latter being an important predictor of susceptibility to parasite infection [44, 73, 74], evidence of a relationship between number of supertypes and peptide repertoire size is still missing and is therefore a contentious issue [75]. Indeed in our own study, the number of MHCI supertypes was not correlated with MHCI sequence divergence (S4 Fig), suggesting that the number of MHCI supertypes may be a poor proxy of functional diversity. The strong effect of MHCI sequence divergence on microbiome composition contradicted our expectations since MHCII is known to have a higher affinity for extracellular peptides than MHCI. However, in our system, MHCI (median of 7 alleles per individual) was much more diverse than the single DRB MHCII locus (a maximum of 2 alleles per individual), thus among individuals variation is much higher in MHCI than MHCII potentially explaining why variation in MHCI was a stronger predictor of microbiome composition than MHCII. Previous work has demonstrated a link between natural diversity at MHCII genes and the microbiome [12]. However, the mechanisms through which the MHCI may regulate the gut microbiome remain largely unknown.

We hypothesized that specific MHC motifs and high functional diversity would selectively target opportunistic pathogens over commensals [14]. However, we did not find a consistent trend across MHC motifs and functional diversity metrics which predicted both positively and negatively the relative abundance of gut commensals and potentially harmful bacteria. Among the former, for instance, we found an increase in the relative abundance of ASVs assigned to *Odoribacter*, a short-chain fatty acid (SCFA)-producing microbe [49], according to a specific MHCI motif (ST*2). SCFA contributes to the integrity of the epithelial barrier and is an important source of energy [76, 77]. Another SCFA generating microbe, Prevotellaceae UCG-001 [78], was both negatively (ST*8 and Migr-DRB*29) and positively (ST*1) associated with specific motifs of MHCI and MHCII. Among potentially harmful bacteria, we observed that a specific MHCII allele (Migr-DRB*36) limited the relative abundance of *Collinsella* and *Campylobacter*, bacteria known to induce inflammatory immune responses by negatively affecting gut permeability [79–81]. In contrast *Campylobacter* was enriched in the presence of the MHCI motif ST*3 and with increasing MHCI functional diversity ($MHCI_{nST}$).

Overall, our results highlight three broad features of the effect of MHC diversity on the microbiome. First, both MHCI and MHCII motifs may play a role in selectively sculpting the microbiome. Second, antigen-specific immune responses driven by the MHC are likely to influence the establishment of a few select microbial taxa (commensals and pathogens) rather than overarching control of the whole microbial community supporting previous experimental findings in germ-free mice by Khan et al. [14]. Third, a restricted number of MHC motifs (in the case of mouse lemurs a single MHCII allele, Migr-DRB*36, that is associated with the relative abundance of 22 ASVs) can significantly shape the microbial flora, a result that is in agreement with the strong HLA-disease associations attributed to bacterial pathogenesis in humans [82–84].

An additional important factor to bear in mind is that there are a number of immune and cell signaling genes linked to MHCI and/or MHCII genes that might contribute to the relationships discussed above. For instance, recent work by Kamitaki *et. al.* [85] demonstrated

that polymorphisms of the C4 protein of the complement system, which is linked to classical HLA genes, is a strong predictor of sex-biased vulnerability to three human illnesses (the auto-immune diseases systemic lupus erythematosus (SLE), Sjögren's syndrome, and schizophrenia), highlighting the need to explore other genes within the MHC in association studies. Hence, future research on the role of adaptive immunity in sculpting the gut microbiome should incorporate both MHCI and MHCII genes, as well as other genes in linkage disequilibrium across the MHC.

A main objective of this study was to evaluate the indirect effect of the MHC on parasite infection via the microbiome. Our data indicates that two taxa are particularly strongly associated with AdV resistance: *Odoribacter*, an SCFA producing microbe, and *Campylobacter*, an opportunistic pathogen. As mentioned above the relative abundance of these microbial bio-markers of AdV resistance is linked to MHC diversity. The enrichment of SCFA generating microbes associated with specific MHC motifs points at the potential involvement of the immunoregulatory effects of SCFA in promoting tolerance. A key mechanism of T-cell lineage differentiation involves SCFAs which are known to shift the balance towards regulatory Foxp3+ T-cells (Treg), limiting inflammation [86–88]. A Treg dominated environment might also be beneficial for opportunistic pathogens, which could explain the observed co-existence between beneficial and potentially harmful bacteria despite high MHC diversity. Pathogenic bacteria can also develop species-specific adaptations to ensure their colonization success [89]. *Campylobacter* uses specialized organelles for adherence and penetration of epithelial cells [90], and *Campylobacter*-specific IgA coating enables *C. jejuni* to aggregate in high densities within the mucus layer, preventing other bacteria from colonizing the gut [91, 92]. There are two possible explanations for the antagonistic interaction between AdV and *Campylobacter*: competitive exclusion and indirect interaction mediated by the immune system. Indeed, both AdV, which is a common persistent infection agent of gut tissue [93], and gut bacterial flora share a common niche. *Campylobacter*-enriched metabolic pathways among AdV^neg individuals supports an antagonistic interaction between *Campylobacter* and AdV infection. Alternatively, infection with either of these potential pathogens is likely to lead to inflammatory immune responses, which will have a knock-on effect on the colonization success of non-targeted parasites.

We found that the bacterial community among helminth infected individuals was more similar than the communities among non-infected individuals. One possibility is that individuals that are free of helminths may be infected with various other pathogens, each leading to a different microbial community. An alternative but not mutually exclusive hypothesis is that helminth infection is shaping the immunological environment of the gut resulting in similar microbial communities among helminth infected individuals. The latter is in agreement with previous work demonstrating that helminth infection selects for bacterial taxa tolerant to a Th2 immunological environment [94]. We found an increase in abundance of Prevotellaceae UCG-001, a SCFA generating microbe [78] associated to multiple MHC motifs, among helminth^pos individuals, further supporting the potential involvement of the microbiome-helminth interaction promoting tolerance via the expansion of Treg cells. Helminths are also capable of metabolizing SCFA, and it has been previously suggested that a mutualistic interaction between commensals and helminths promotes gut homeostasis [94]. This phenomenon known as the "hygiene-hypothesis" has gained considerable support from epidemiological studies [95, 96] and suggests that helminth infection in mouse lemurs may influence immune responses.

Experimental studies using model organisms have been crucial to our understanding of the mechanisms that mediate host-microbe interactions [13, 14, 82, 97]. However, gut microbiomes of lab animals have been shown to be unrepresentative of natural populations [98, 99]

and provide limited scope of furthering our understanding of the interplay between the natural diversity of MHC genes, the microbiome, and parasite infection. Wildlife studies therefore represent an important complement to experimental approaches but are lacking. Here, we aimed at using a non-invasive approach in a natural primate population to test the hypothesis that MHC diversity influence parasite infection directly and indirectly via the microbiome. Nonetheless, like most wildlife studies, our work offers a snapshot perspective of ongoing parasitic challenges with no knowledge of exposure to parasites throughout an individual's lifetime. Previous infections can modify the immune environment of the host [100, 101], for instance by affecting the Th1-Th2 balance [102, 103] or by shifting the availability of microbial metabolic compounds [104, 105], with a knock-on effect on susceptibility to infection. Moreover, a history of infection co-occurrence in helminth-mice models have been shown to influence immunity to invading parasites even after helminth clearance [102, 106]. Therefore, our understanding of the interrelationship between natural adaptive genetic diversity, parasites, and the microbiome would benefit from longitudinal and long-term studies with repeated sampling and profiling of microbial communities and parasitic challenges.

## Conclusion

Our correlative study suggests that the preferential establishment of specific bacterial taxa in a MHC-dependent manner shapes the microbial environment and functional profiles of the bacterial community, potentially influencing the ability of invading parasites to successfully establish an infection. Under this scenario, the host benefits from the antagonistic interactions between specific microbes and parasites, thus providing evidence of an indirect link between MHC diversity and parasite resistance via the microbiome. Our understanding of the extraordinary diversity of MHC genes has been focused on parasite-mediated mechanisms, however our findings highlight the potential role of microbiome-driven selection as yet another layer involved in the co-evolutionary dynamics acting on MHC diversity.

## Materials and methods

### Ethics statement

Our work was approved by the ethics committee of the Institute of Zoology of Hamburg University, the University of Antananarivo and Madagascar National Parks. Approval was granted by the Autorisation de Recherche No. 54/13/MEF/SG/DGF/DCB.SAP/SCB of February 22, 2013, issued by the Direction Générale des Forêts and the Direction de la Conservation de la Biodiversité et du Système des Aires Protégées of the Ministère de l'Environnement, et des Forêts and exported to Germany under the CITES permit 576C-EA09/MG14. All animals were handled in accordance with the relevant guidelines and regulations.

### Data collection and parasite screening

We live-trapped *M. griseorufus* at two sites in the Mahafaly Plateau in southwestern Madagascar. Both sites show a semi-arid climate, characterized by irregular rainfall that increases from west to east [107, 108] but differ in the degree of anthropogenic disturbance. While one site, Andranovao (24˚01′S; 43˚44′E) is located in the Tsimanampetsotsa National Park and consists of intact dry spiny forest, the other one is in Miarintsoa (23˚50′S; 44˚6′E), a village approx. 40 km east from the National Park consists of open anthropogenic landscape and degraded forest fragments. Sampling was done during the dry seasons of 2013 and 2014 and the wet seasons of 2014 and 2015. We live-trapped *M. griseorufus* using Sherman traps baited with bananas (for details see [109]). All animals were individually marked with electronic transponders. After

processing of animals and sample collection, we thoroughly cleaned all traps before using them again. Samples were stored at ambient temperature for a few days to weeks in the field, then for variable length of time in a freezer before being transported to Germany where we kept them at -20 C until DNA extraction. We collected small ear biopsies from 143 anesthetized animals and preserved them in 90% ethanol for later MHC characterization. When possible, we collected fecal samples from each trap or handling bag and preserved one subsample in 500 ul RNAlater (Life Technologies) for later adenovirus (AdV) screening ($n_{Total}$ = 143; $n_{Andranovao}$ = 101; $n_{Miarintsoa}$ = 42) and microbial sequencing and stored another subsample in 70% ethanol for parasitological analysis ($n_{Total}$ = 85; $n_{Andranovao}$ = 57; $n_{Miarintsoa}$ = 28).

## AdV and gastrointestinal parasite screening

We used an Illumina MiSeq platform and followed a target-specific semi-nested PCR assay to evaluate AdV prevalence in 143 mouse lemurs as previously described by Wasimuddin et al. [35]. We counted helminth eggs of a subset of 85 mouse lemurs by using a McMaster flotation technique and a potassium iodide solution [57, 110]. A major concern among epidemological field studies is the low sensitivity of current methods to provide estimates of parasite burden and diversity [111, 112], particularly in wildlife [113]. We therefore use overall helminth infection status as a conservative proxy of gastrointestinal parasite infection. We entered AdV and helminth infection status as presence / absence data for statistical analyses. We emphasize that our data reflects punctual infection status rather persistent infection.

## Microbiome sequencing and data processing

Sequencing data of the 16S rRNA gene of the hypervariable V4 region stems from a previous study demonstrating an effect of AdV on the gut microbiome of *M. griseorufus* [35]. The curated dataset used in this study consists of 6,521,596 reads, with a mean coverage of 45,605 reads (min = 18950, max = 109156) per sample. We processed the microbiome sequencing data with the QIIME2 pipeline [114] and used the implemented DADA2 algorithm [115] for data denoising, merging and calling of ASVs. For taxonomic assignment we used the Silva database [116]. We excluded all sequences that could not be assigned to a bacterial taxa at the phylum level. We imported and further analysed the microbiome sequencing data output from QIIME2 in the R environment using the PhyloSeq package [117]. We computed microbiome diversity (Faith's phylogenetic diversity and Shannon's diversity index) and estimated divergence in microbiome composition (mean distance from the centroid) using Bray-Curtis distance matrix across a 10000 bootstrap sample. For analyses of diversity and composition diversity, we filtered out rare ASVs using a prevalence threshold of <0.1 and <0.2, respectively. Out of the 1505 ASVs identified across the 143 individuals, following prevalence filtering we retained 169 for analyses of diversity and 116 ASVs for analyses of microbial composition.

We predicted functional profiles of the 16S rRNA gene sequencing data at the metabolic pathway level (MetaCyc) by using the PICRUSt2 pipeline with the default parameters [50]. We exported the stratified pathway abundances and taxonomic contributions for analyses in Songbird [45] and in R. The same filtering described above for the taxonomic features was applied for analyses of diversity and composition of the pathway features.

## MHC characterization and diversity estimates

We used high-throughput amplicon sequencing to characterize the MHC class I region (MHCI), MHC class II exon 2 DRB gene (MHCII) on an Illumina MiSeq platform. For MHC characterization, we isolated genomic DNA from ear biopsies by using the Qiagen DNneasy

Blood & Tissue Kit (Qiagen). We designed target specific primers to amplify the MHCI gene (fragment length = 236 bp, up to 10 loci; MHCI-Migr-F: 5′-CCCAGGCTCCCACTCCCT-3′ and MHCI-Migr-R: 5′-GCGTCGCTCTGGTTGTAGT-3′). We also used target specific primers [118] to amplify the MHCII-DRB gene (fragment length = 171 bp, 1 locus, JS1:5′-GAGTGTCATTTCTACAACGGGACG-3′ and JS2 5′-TCCCGTAGTTGTGTCTGCA-3′). DQA, DQB and DRB MHCII genes in mouse lemurs are tightly linked [119, 120]. We therefore focus on the MHCII-DRB gene since previous work identified a larger number of sites under positive selection for this gene compared to the DQB [120] and tight linkage would have limited our ability to distinguish effects between MHCII genes. Both MHCI and MHCII genes were previously described as functional regions coding for antigen binding sites [118, 119, 121].

We prepared the Illumina sequencing libraries by performing two consecutive rounds of PCR following the approach of Fluidgm System (Access Array System for Illumina Sequencing Systems; Fluidigm Corporation). We sequenced the libraries using the Illumina MiSeq platform. Sequencing was done using technical replicates for 85% and 100% of the samples at the MHC class II-DRB and MHC class I gene respectively. Sequence variants of the MHCI gene were identified as true alleles if they were identified in both technical replicates. Negative controls over sequencing runs were clean (< 50 reads after merging). We generated a total of 15,050,630 and 749,428 paired-end reads to characterize allelic diversity at the MHCI genes (average of 51,192 reads, min = 6031, max = 319259) and MHCII gene (average of 2,306 reads, min = 392, max = 3649), respectively.

Sequence data was processed using the ACACIA pipeline [122] (code available under https://gitlab.com/psc_santos/ACACIA) using a proportion threshold (low-por) of 0.01 and 0.10 for allele calling on the MHCI and MHCII-DRB dataset, respectively. Only individuals that exhibited 100% repeatability in allele calling between both sets of replicates of the highly duplicated MHCI gene were kept in downstream analyses which resulted in the exclusion of 2 individuals). Using a generalized linear mixed model (GLMM) that controlled for replication across individuals, we found no statistical support for a relationship between sequencing depth and the number of MHC alleles (S11 Fig), suggesting that sequencing depth was sufficient for reliable genotyping in our system. MHCI is a highly variable region characterized by a series of duplications [119, 121], while the MHCII DRB gene is non-duplicated in mouse lemurs [118, 120, 123]. In contrast to MHCII, the influence of diversity at the MHCI region on parasite resistance remains poorly explored. MHCI and II genes are not linked in mouse lemurs [119] and we therefore expect them to exhibit functional dissimilarities.

We aimed to quantify functional MHC diversity at each MHC gene. To this end we first identified the sites under positive selection (PSS) using the maximum likelihood analysis implemented in codeML (PAML software [124]), since these sites are likely to be protein binding sites that recognise antigens. Positive selection is indicated by $d_N/d_S$ ratio ($\omega$) > 1. The following models of codon evolution were computed: M7 (assumes variation of $\beta$: $\omega$ among codons modelled under a $\beta$ distribution and does not allow for positive selected sites) and M8 (similar to M7 but assumes $\omega$ > 1). Model M7 serves as a null model and can be compared to model M8 by means of the likelihood-ratio test. To identify the best fitting model the twice log likelihood difference is compared with a $\chi^2$ distribution. Subsequently, if the model indicating selection (M7) results in a significant better fit to the data, the Bayesian approach in CODEML was used to determine the identity of sites under positive selection. Thirteen and eight positively selected sites were identified in the allele sequences of the MHC I and MHC II genes, respectively (S1 Table). For the MHCI, PSS amino-acid sequences of each allele were described by physicochemical attributes (z-values), and we used a discriminant analysis of principal components (DAPC, R package adegenet [125]) to group MHCI alleles into clusters

(supertypes). Given that the MHCII DRB is a non-duplicated gene, we considered functional MHC variants that consist of unique amino-acid sequences (49 out of 51 alleles) at PSS and for statistical analyses, we included alleles present in more than three individuals (frequency > 0.03) (S1 Fig). We used the number of MHCI supertypes ($MHCI_{nST}$) per individual as a measure of individual diversity. Since there is only a single locus at the MHCII and 97% of individuals had the maximum of two alleles at this gene, the number of MHCII supertypes was not estimated. In order to assess sequence diversity (divergence of alleles within an individual [126]), we estimated mean sequence divergence over positive selected sites (distPSS) based on the Grantham distance matrix [127] as done in [73] for both the MHCI ($MHCI_{distPSS}$) and the MHCII-DRB ($MHCII_{distPSS}$). This approach enabled us to take into account the physicochemical properties of the PSS amino-acid sequences, being an improved proxy of MHC functional divergence. Measures of MHC diversity were not correlated (S4 Fig).

## Statistical analyses

We used odds-ratio tests to evaluate the relative risk of being infected with AdV or helminths among individuals carrying a specific MHC motif. We corrected $P$-values for multiple comparisons across MHC motifs using the Bonferroni correction.

To assess the link of MHC diversity, microbiome diversity and the covariates site, season and sex on infection status, we fitted GLMs with a binomial error structure. Note that we did not include age as a covariate in our models since age is notoriously difficult to estimate accurately in mouse lemurs since young and older adults cannot be reliably distinguished by their outer appearance [128]. Weight can be used as a proxy of reproductive state [35] however in our study it is collinear with season. We therefore prioritised controlling for season over age. We ranked the models using the information-theoretic (I-T) model selection procedures, with the Akaike information criterion adjusted for small sample sizes (AICc) with the MuMIn package [129]. We used model averaging (without shrinkage) across the credible set of models (cumulative AICc weight < 0.95) to generate parameter estimates. We interpreted the effect of variables on our response variable if 95% confidence intervals did not overlap with zero. We report effect sizes using partial-r for continuous variables and Cohen's D for categorical variables and estimate 95% confidence intervals by boostrap (n = 10000) [130].

We fitted linear term for $MHCI_{nST}$, $MHCI_{distPSS}$, and $MHCII_{distPSS}$ and covariates in a GLM with a gamma distribution to test for the association between MHC functional diversity and alpha diversity of the microbiome. We used model selection and 95% confidence intervals to interpret the effect of these predictors on microbiome alpha diversity.

We explored the contribution of the predictor variables on the bacterial (taxa) and functional (metabolic pathways) community structure by using canonical correspondence analysis (CCA) using the vegan package in R [131] on a chord transformed matrix. We tested for significance of the overall CCA model by means of permutation (9999). We used the *envfit()* function to assess the significance of the fitted vectors (MHC functional diversity) and factors (MHC specific motifs, site, season, sex, AdV and helminth infection). The effect of each MHC motif was assessed independently in separate models along with all other covariates.

To explore the effects of MHC class I and class II motifs on the relative differential abundance of specific ASVs, we applied an analysis of composition of microbes (ANCOM) [132] which controlled for the effect of site and season. ANCOM runs a set of pairwise tests for each ASVi and across each ASVj, where the null sub-hypothesis is that the log ratio ASVi/ASVj is not associated with a given predictor (the total number of sub-hypothesis is the total number of ASVs investigated minus one). A *W* score is the sum of rejected null sub-hypotheses. Therefore, *W* estimates the strength of support for a relative differential abundance of a specific ASV

according to a predictor of interest. We used volcano plots to visualize the relationship between the $W$ score and the estimates of the differential logs from a linear model. We used a $W$ score threshold of 0.6, which is equivalent to 60% null sub-hypotheses rejected across the set of pairwise tests, and a $P$ value $< 0.005$ to identify ASVs whose relative abundance differs a according to the presence of specific MHC motif. We used differential ranking [45] to identify ASVs most associated with low, and high MHC diversity by means of a multinomial regression model with $MHCI_{distPSS}$, $MHCI_{nST}$ and $MHCII_{distPSS}$, site and season as covariates, using Songbird [45]. We used a batch size of 5, epochs of 50,000 and a prior of 0.8 as parameters for the analysis. The coefficients of the multinomial regression represent the log fold change for each ASV along the range of MHC functional diversity estimates.

In order to investigate whether microbial taxa associated to specific MHC motifs and MHC functional diversity contribute to AdV and helminth infection status, we used a reference frame approach. Traditional methods of compositional analyses do not allow to make inter-pretations of absolute differences in ASV abundance. However, by using a carefully chosen reference that is not expected to differ in absolute loads across covariates and is known to be stable across environments, the reference frame allows us to overcome the limitations associ-ated with interpreting differences in relative abundance and to make some cautious inference about absolute differences [45]. To calculate log-ratios, we defined *Bifidobacterium* spp. as a reference frame given that: i) it exhibits high prevalence and, ii) we expect this taxa to be rela-tively stable within the microbial community because a series of strategies that enable host tol-erance have been identified among species of bifidobacteria. Indeed, previous studies have demonstrated that immuno-modulatory roles of bifidobacteria can lead to reduced inflamma-tion [133, 134] and the production of key molecules enable bifidobacteria to withstand stress-ful environmental conditions within the gut [23, 135], making this taxon a useful reference of health. Here we report the differential abundance of taxa according to AdV and helminth infection relative to the abundance of *Bifidobacterium* spp. using a Mann-Whitney Wilcoxon test. We considered core microbial taxa (with a prevalence $> 60\%$) that were ranked high or low along this gradient as suitable candidates for assessing (with a log-ratio test) if infection status influences their relative abundance. We controlled for multiple comparisons across taxa by means of Bonferroni correction of $P$-values. We report estimates of effect size (Cohen's D and 95% confidence intervals). In order to be conservative, we chose to interpret only effects with a medium to large effect $> |0.50|$ and whose 95% confidence intervals do not overlap 0.

We also used ANCOM and differential ranking to identify the metabolic pathways associ-ated with MHC motifs and functional diversity, respectively, following the same approach used for taxonomic based analyses. We then used the subset of pathways linked to MHC diver-sity to test whether they predict AdV and helminth infection status. We used with $MHCI_{distPSS}$, $MHCI_{nST}$ and $MHCII_{distPSS}$, site, and season as covariates in the multinomial regression model with a batch size of 5, epochs of 50,000 and a prior of 1 as parameters for the analysis using Songbird. To calculate log-ratios we used the TCA pathway (PWY-7254) as a reference since this is a common pathway central to the generation of energy and present in most aero-bic living organisms.

## Supporting information

**S1 Fig. Clustering of MHC class I alleles into supertypes.** Clustering of MHC class I alleles into supertypes using discriminant analysis of principle components (DAPC) based on a matrix of physiochemical properties of the sites under positive selection (PSS) of each allele. *(A)* Number of clusters chosen using the find.clusters() function of the adegenet package

[125]. The red dashed line shows *(k)* = 17. *(B)* Scatterplot of the first and second discriminant functions showing the supertype clusters.
(TIF)

**S2 Fig. MHC class II DRB allele frequency.** Relative frequencies of MHC class II DRB alleles in *Microcebus griseorufus* (n = 143). The identity of allele variants sharing identical amino −acid sequences at positive selected sites (PSS) is shown in parenthesis. Alleles Migr-DRB*28 to Migr-DRB*50(*58) were present in more than 3 individuals and the remaining rare alleles were not included in subsequent analyses.
(TIF)

**S3 Fig. Association between specific MHC motifs and AdV and helminth infection status.** Relative motif frequencies of MHCI supertypes *(A)* and MHCII alleles found in more than 3 individuals *(B)*. Frequency of MHCI supertypes according to AdV *(C)* and helminth infection status *(D)*. Frequency of MHCII alleles according to AdV *(E)* and helminth infection status *(F)*.
(TIF)

**S4 Fig. Correlation plot between different measures of MHC diversity.** Values within the top diagonal boxes of the correlation plot correspond to Spearman correlation coefficients.
(TIF)

**S5 Fig. Canonical correspondence analysis (CCA) biplot for taxonomic composition in association with MHC functional diversity.** Canonical correspondence analysis (CCA) for taxonomic composition in association with MHC functional diversity (n = 143). CCA biplot depicting ASVs (full circles) and MHC diversity estimates as arrows. $MHCI_{distPSS}$ ($r^2$ = 0.16, $P$<0.0001) functional diversity significantly predicting microbiome composition is shown in red. Labels correspond to assigned taxa.
(TIF)

**S6 Fig. CCA biplot depicting the differences among samples according to MHCI super-types.** CCA biplot depicting the differences among samples according to the presence/absence of the MHCI supertypes.
(TIF)

**S7 Fig. CCA biplot depicting the differences among samples according to MHCII alleles.** CCA biplot depicting the differences among samples according to the presence/absence of the MHCII alleles.
(TIF)

**S8 Fig. CCA biplot depicting the compositional differences of functional pathways among samples according to AdV and helminth infection status and covariates.**
(TIF)

**S9 Fig. The effect of covariates on functional pathway diversity and composition.** Boxplots illustrating differences in pathway eveness according to season *(A)*, pathway richness according to site *(B)* and CCA biplot depicting differences among samples according to season *(C)*.
(TIF)

**S10 Fig. CCA biplot depicting the differences among samples according to MHCI and MHCII motifs.** CCA biplot depicting the differences among samples according to the presence/absence of the MHCI supertypes and MHCII alleles.
(TIF)

**S11 Fig. Relationship between sequencing depth and number of alleles for MHCI and MHCII genes.** Association between sequencing depth and number of MHCI alleles *(A)*, after removal of an outlier *(B)*, and the number of MHCII alleles *(C)*. Sequencing depth used for these analyses was after denoising, i.e. after removal of chimeras, singletons and non-target sequences according to a BLAST search using the ACACIA pipeline (see Material and methods for pipeline methods). Generalized linear mixed models (GLMM) for MHCI *((A B))* included number of alleles as a response variable, sequencing depth as an explanatory variable and sample ID as a random factor and were fitted with a Poisson distribution with a log link. The GLMM for MHCII *((C))* included number of alleles as a response variable (1 or 2 alleles), sequencing depth as an explanatory variable and sample ID as a random factor and was fitted with a Binomial distribution with a logit link.
(TIF)

**S1 Table. Polymorphic sites exhibiting signals of positive selection.** Log-likelihood values and parameter estimates of models testing for positive selection acting on MHC class I and MHC class II DRB exon 2 of *Microcebus griseorufus*. Parameters $p$ and $q$ computed from the beta distribution. $\omega = d_N/d_S$ ratio. $p_n$ = of sites that fall into $\omega n$ site class. Site positions inferred to be under positive selection estimated at a * 95% and ** 99% confidence interval level. Amino acid numbering is based on the HLA-A2 haplotype.
(TIF)

**S2 Table. Top ranking GLMs examining the association between MHC, microbiome diversity (Faith's PD), and covariates on AdV and helminth infection status.** Model selection of GLMs (top 20 models subset) examining the association between MHCI and MHCII motifs and diversity estimates, microbiome diversity (Faiths' PD) and divergence (Bray-Curtis) and covariates on AdV and helminth infection status. $AIC_c$ = Akaike Information Criterion for small sample sizes; $AIC_c\omega$ = AIC weight; "+" denotes categorical parameters included in the model whilst the inclusion of a continuous variables is indicated with its estimate.
(TIF)

**S3 Table. Top ranking GLMs examining the association between MHC, microbiome diversity (Shannon's index), and covariates on AdV and helminth infection status.** Model selection of GLMs (top 20 models subset) examining the association between MHCI and MHCII motifs and diversity estimates, microbiome diversity (Shannon's index) and divergence (Bray-Curtis) and covariates on AdV and helminth infection status. $AIC_c$ = Akaike Information Criterion for small sample sizes; $AIC_c\omega$ = AIC weight; "+" denotes categorical parameters included in the model whilst the inclusion of a continuous variables is indicated with its estimate.
(TIF)

**S4 Table. Model average parameter estimates and 95% confidence intervals of GLMs examining the association between MHCI and MHCII diversity estimates, microbiome diversity (Faiths' PD) and divergence (Bray−Curtis) and covariates on AdV and helminth infection status.**
(TIF)

**S5 Table. Model average parameter estimates and 95% confidence intervals of GLMs examining the association between MHCI and MHCII diversity estimates, microbiome diversity (Shannon's index) and divergence (Bray−Curtis) and covariates on AdV and helminth infection status.**
(TIF)

**S6 Table. Model average parameter estimates and 95% confidence intervals of GLMs examining the association between phylogenetic diversity (Faiths' PD) and evenness (Shannon's diversity index), MHCI and MHCII functional diversity, and covariates.**
(TIF)

**S7 Table. Model average parameter estimates and 95% confidence intervals of GLMs examining the association between predicted pathway richness and evenness (Shannon's diversity index), MHCI and MHCII functional diversity, AdV infection status, and covariates.**
(TIF)

**S8 Table. Model average parameter estimates and 95% confidence intervals of GLMs examining the association between predicted pathway richness and evenness (Shannon's diversity index), MHCI and MHCII functional diversity, helminth infection status, and covariates.**
(TIF)

## Acknowledgments

The study was carried out under the partnership agreement between MNP (Madagascar National Parks), the Department of Animal Biology, University of Antananarivo and the Department of Animal Ecology and Conservation, University of Hamburg. We are very grateful for the help and the logistic support provided by all the field assistants, and Kerstin Wilhelm and Ulrike Stehle for their support in the lab.

## Author Contributions

**Conceptualization:** B. Karina Montero, Wasimuddin, Nina Schwensow, Mark A. F. Gillingham, Victor Corman, Jörg U. Ganzhorn, Simone Sommer.

**Data curation:** B. Karina Montero, Yedidya R. Ratovonamana, S. Jacques Rakotondranary, Victor Corman.

**Formal analysis:** B. Karina Montero, Nina Schwensow, Mark A. F. Gillingham.

**Funding acquisition:** Jörg U. Ganzhorn.

**Investigation:** B. Karina Montero, Wasimuddin, Yedidya R. Ratovonamana, S. Jacques Rakotondranary, Victor Corman.

**Methodology:** B. Karina Montero, Nina Schwensow, Mark A. F. Gillingham.

**Project administration:** Victor Corman, Jörg U. Ganzhorn, Simone Sommer.

**Resources:** Mark A. F. Gillingham, Christian Drosten, Jörg U. Ganzhorn, Simone Sommer.

**Software:** B. Karina Montero, Mark A. F. Gillingham.

**Supervision:** Jörg U. Ganzhorn, Simone Sommer.

**Visualization:** B. Karina Montero, Mark A. F. Gillingham.

**Writing – original draft:** B. Karina Montero.

**Writing – review & editing:** B. Karina Montero, Nina Schwensow, Mark A. F. Gillingham, Jörg U. Ganzhorn, Simone Sommer.

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
