## [Decision Letter · Decision Letter 0]

29 Jun 2021

Dear Dr. Montero,

Thank you very much for submitting your manuscript "Evidence of MHC class I and II influencing viral and helminth infection via the microbiome in a non-human primate" for consideration at PLOS Pathogens. As with all papers reviewed by the journal, your manuscript was reviewed by members of the editorial board and by several independent reviewers. In light of the reviews (below this email), we would like to invite the resubmission of a significantly-revised version that takes into account the reviewers' comments.

The reviewers all found the work to be of interest, but raised several concerns that need to be addressed. The authors are encouraged to address all of the reviewers' concerns and resubmit their manuscript with a point-by-point response.

We cannot make any decision about publication until we have seen the revised manuscript and your response to the reviewers' comments. Your revised manuscript is also likely to be sent to reviewers for further evaluation.

Sincerely,

Jason M. Brenchley

Guest Editor

PLOS Pathogens

P'ng Loke

Section Editor

PLOS Pathogens

Kasturi Haldar

Editor-in-Chief

PLOS Pathogens

orcid.org/0000-0001-5065-158X

Michael Malim

Editor-in-Chief

PLOS Pathogens

orcid.org/0000-0002-7699-2064

The reviewers all found the work to be of interest, but raised several concerns that need to be addressed. The authors are encouraged to address all of the reviewers' concerns and resubmit their manuscript with a point-by-point response.

Reviewer's Responses to Questions

**Part I - Summary**

Reviewer #1: The study focuses on an area of tremendous interest; the contribution of host genetic variability, particularly within the major histocompatibility complex, in shaping the composition of the gut microbial community. The premise is particularly important and the authors combine modern laboratory techniques with field work, which is both novel and highly interesting.

However, I have serious concerns regarding the interpretation and presentation of the data (e.g., an animal is either AdV+ or AdV-; your assay either detected AdV or it didn't. There is no reason to present a per-individual probability of infection stratified by MHC status. Either the MHC group is enriched for AdV+ or AdV- individuals or it isn't.). All such oddities need to be corrected and better communicated. No modeling is necessary to test the hypothesis that AdV-positivity segregates with DRB*34-positivity. Keep it simple.

Samples were collected from the same species of lemur at two distinct sites and at different time periods (i.e., different seasons), which differ significantly with regard to variety of parameters that could influence the composition of the gut flora. More critically, the populations at the different sites may be genetically distinct, but there is no presentation of data or discussion of results that indicate that the authors thoughtfully partitioned their datasets to determine if their analyses were biased by site (I suspect that they are, but how could I know– they only state “we evaluated… while controlling for extrinsic factors (sampling site and season)” and “we used binomial GLMs to examine the effect of MHC and microbiome diversity on parasite infection while controlling for the potential effects of covariates (site, season, and sex).” That is pretty all encompassing. I want to see the raw data.

The authors assume that once infected, AdV infection is persistent. Yet, there is no discussion of this reality that AdV infection can be transient and therefore, particularly without any immunological data, it is impossible to know the true relationship between host genetics and susceptibility to infection. The authors clearly have the ability to address this, and so they should.

The authors need to show that the MHC composition gut flora does not differ by site if they are going to combine the two populations of animals.

This manuscript needs significant revision; not necessarily new work, but potentially re-analysis to simplify the substantiation of the authors' largest claim. It is otherwise an important topic that few groups are positioned to investigate, and this group is clearly well positioned.

Reviewer #2: This manuscript describes a project to correlate information about MHC class I and class II sequences, bacteria in the intestinal microbiome, helminth carriage and adenovirus infection using a 124 individuals of a wild mouse lemur colony. The data collection alone must have been an enormous effort, and then a wide variety of statistical analyses were applied to look for correlations. I am not qualified to evaluate the model building and statistics, but I know a fair amount about the MHC, and a bit about viruses, bacteria, helminth parasites and immune responses. Overall, the authors are to be commended on an outstanding paper with very sensible approaches, but I do have a few concerns which the authors might do well to consider.

Perhaps my major concern is the assumption that the MHC class I and class II genes that are typed by sequencing and used for the correlations are responsible for all the biological effects. The MHC in general is a large region with many different kinds of genes, and there are many examples of traits initially ascribed to classical class I and class II genes that turned out later to be due to linked genes with entirely different functions. Just one very recent example is the importance of C4 alleles on sex-linked diseases including various autoimmune syndromes and schizophrenia (Kamitaki et al 2020 Nature). However, it is not just the genes involved in immunity such as the complement components factor B, C2 and C4 and the cytokines TNF and lymphotoxins, but genes that might affect any aspect of physiology, cell cycle and the like, given that it is complex biology being examined. It is easy to imagine a cell cycle or TRIM gene (located near a particular class II allele) with a point mutation in the coding region or in the promoter/enhancer to affect adenovirus resistance. It is even plausible that all the class I genes of a particular supertype have a common ancestor, so that the surrounding haplotypes of all those class I alleles contain an allele of C4, TNF or steroid hydroxylase that affects helminths. The careful interpretations and discussions throughout the manuscript do not need to be re-written, but near the beginning and at the end, the authors should consider these alternatives, which afford a wider view of the possibilities.

A second concern is that the authors have chosen to lump class I alleles together as supertypes. I am aware of the long and laudable history of simplifying the complexities of human MHC alleles by combining closely-related alleles together, the need to reduce the number of alleles to achieve significance in epidemiology, and the fact that there are no data about actual peptide binding for most non-model organisms. However, there are ample data to show that closely-related alleles can have enormously different effects: consider the many subtypes of HLA-B27 for ankylosing spondylitis (predisposing *2705 and protective *2709 differ by a single amino acid!), or the subtypes of HLA-DRB1*04 which are either strongly predisposing or protective for rheumatoid arthritis (protective *0402 and predisposing *0401/*0404 differ by three amino acids, one of which is key). Moreover, supertypes do not correlate with the size of peptide repertoire (see for instance Kaufman 2021 Frontiers Immunol), which has recently emerged as an important determinant of resistance and susceptibility to infectious pathogens (summarised in one of the papers cited by the authors, Kaufman 2018 Trends Immunol, and given further support for HIV progression by Bashirova et al 2020 PNAS). Again, the authors do not necessarily need additional experiments or extensive revisions, but these issues should be discussed.

A third concern is that the authors have apparently analysed a wide variety of class I genes, but focused only on the DRB family of class II genes. I could find at least one other pair of class II genes in the mouse lemur genome on ENSEMBL (likely to be DQA and DQB). What is the evidence that other kinds of class II genes (DQ-like and perhaps DP-like) are not important? Again, this needs to be discussed.

A final major concern is the provision of data, rather than just presenting the interpretations of statistical analyses. Unless the authors have a strong reason for withholding such information, the sequences of the alleles (along with their assignment to supertypes for class I) is crucial for readers to understand the paper in more than a superficial way. Also, it is important to include a list of all the individuals examined, with the assignments of MHC alleles along with the presence of adenovirus, helminths and the various bacterial species in the microbiome. The data for the adenovirus and helminths could be presented as presence/absence, or better as MiSeq reads for adenovirus and egg counts for helminths. All this data could be easily presented as a few spreadsheets in supplementary files. The provision of original data is now routine for many biomedical journals and clinical studies, which allows others to examine the same data for validation and for additional research, adding much value to the original paper.

Reviewer #3: In this article Montero and colleagues investigate the links between diversity at MHC loci (class I and II), gut microbiome composition and infection by gut parasites (adenoviruses and helminths) in wild mouse lemurs. Importantly, the authors intended their paper to also address the question of potential indirect effects of MHC variation on parasite infection probability that would be mediated by the microbiome. By applying a broad range of statistical analyses they find that (i) a specific MHCII motif is associated with AdV positivity, (ii) MHCI variability as well as specific MHC I and II alleles influence microbiome composition, mostly by their effect on a few ASV abundance, (iii) among the ASV whose abundance was predicted by variation at MHC loci, some exhibited different relative abundance in individuals infected or not by gut parasites, (iv) most of these findings find a reasonable echo in analyses focused on bacterial functional pathways, not taxonomy.

This study addresses a consequential series of questions about the relationships between the immunogenome and the organisms it controls using a tri-partite data set (host, gut microbiome and gut parasites), which is still relatively rare. The paper reads well and I find the results very interesting, although, as acknowledged by the authors in the first part of the discussion, the nature of their study (observational and cross-sectional) prevents truly demonstrating indirect effects of MHC diversity on gut parasites mediated by the gut microbiome. In general, I’d be rather supportive of seeing this article published in PLoS Path. However, given its extensive use of a broad set of statistical tools I would find it reassuring to know it has been checked by a trained statistician (I am not this person). I also have a couple of major and minor remarks that the authors might address in a revision.

Sebastien Calvignac-Spencer

**Part II – Major Issues: Key Experiments Required for Acceptance**

Reviewer #1: Simplify your analyses where possible. This is a major generalization, but the hypotheses is much simpler than is presented: “MHC genotype influences infection susceptibility to X” and “MHC genotype influences the composition of the gut microbial communIty.”

The authors need to show that AdV-negative animals are sero-negative as well and they need to control for animal age. The same needs to be done for Helminth… they make a major assumption that once infected, always infected, and it isn't entirely clear that this is the case.Otherwise, they need to reconstruct their premise.

Reviewer #2: I did not find the need for additional experiments to support this study, only additional presentation (sequences of class I and class II alleles, list of individuals with MHC type along with presence/absence of adenovirus, helminths and various bacterial taxa of the microbiome).

Reviewer #3: Major remarks

1) As afore mentioned the first part of the discussion is very careful about causality. In my view, it makes much sense to be careful since, in essence, their results are compatible with indirect effects mediated by the microbiome, but they are not a formal proof of the chain of causality. However, the authors still use a wording clearly suggestive of causality in multiple occasions throughout the ms. I did not compile a complete list but here a couple of examples, e.g. in the title, in the abstract “evidence of the indirect effect of the MHC via the microbiome” or l. 151.152 “resistance to AdV was provided by an increase in Campylobacter”. It’d be great if the authors could carefully check through the ms that their wording is consistently careful.

2) The entire analysis relies on appropriately identifying MHC alleles. It is a complicated task with which the authors are very familiar. Here they identify many more MHCI than MHCII alleles. It turns out they also sequenced their MHCI amplicons much deeper than their MHCII. I imagine that this was maybe dictated by the structural differences they mention (with multiple paralogs of MHCI existing in mouse lemur genomes where MHCII is a single locus) and I also imagine that the workflow they designed and recently published can control for this, but it’d be good if the authors could show some data showing that these different depths could not explain the different allele numbers (e.g. sub-sampling their MHCI amplicons to the MHCII depth).

**Part III – Minor Issues: Editorial and Data Presentation Modifications**

Reviewer #1: You state that 29.4% of individuals = 143 animals on line 57. You also state that 40.1% of individuals = 86 animals on 57-58. While I understand that you are referring to percentages for subsets, the way it is presented is confusing.

Line 86 and 93, you refer to the wrong figures.

Reviewer #2: Detailed concerns.

Page 3, legend to Fig. 1. “… we predicted that MHC I and MHC II variability influences the diversity and the composition of the microbiome.” I was pleased to see that the authors defined the words motifs, diversity and divergence for MHC (which are used in different ways by different people) in terms of their assays, but in this phrase they introduce the term “variability”. What do they mean by that, and how does it differ from “diversity”, etc? I would be grateful if the authors checked through their text to ensure that the reader can understand what is meant by these sometimes-loaded terms.

Page 3, lines 57-59. “…29.4% of individuals tested AdV+ (n=143), 40.1% of 57 individuals (n=86) were infected with helminths, and 8.6% of individuals were co-infected with AdV and helminths…”. 8.6% double infections is different from the product of the two single infections (11.8%), so I was curious whether this is a significant difference and whether it might be related some kind of exclusion (as mentioned for Campylobacter and adenovirus).

Page 3, lines 59-63. “We detected a total of 226 MHCI functional alleles (median of 7 alleles per individual, range 4-20) in the population and grouped them into 17 MHC supertypes (ST) based on physicochemical variables (z-values [32]) of the amino acid sites affected by positive selection (PSS) (Materials and Methods, S1 Fig and S1 Table).” Having just complimented the authors on defining their terms, “functional allele” is another term with several meanings, but I see that I was pointed to the Materials and Methods, where the authors carefully describe what they mean. Still, it would be helpful to rearrange the sentence (if I have understood correctly) to reflect this: “We detected a total of 226 MHCI functional alleles (median of 7 alleles per individual, range 4-20) in the population based on physicochemical variables (z-values [32]) of the amino acid sites affected by positive selection (PSS) and grouped them into 17 MHC supertypes (ST) (Materials and Methods, S1 Fig and S1 Table).”

Perhaps a more important point is where are the sequences? In this age of supplementary materials, providing the original data on which the analyses are carried out is routine for biomedicine, and should be for this kind of (nearly) unique study. Perhaps the source of the sequences is mentioned somewhere, but I certainly didn’t notice it.

As just one example of the benefits for the reader and the scientific community, I was unable to learn which sites are PSS in the S1 Table, since the basis of the numbering was not mentioned (is it from the start of the coding region including the signal sequence, it is from the presumed start of the mature protein; in both cases are they the same for all alleles detected; is the numbering based on the sequence of HLA-A2 as has been standard for decades or what exactly?).

Page 3, lines 63-65. “For the MHCII, we identified 49 functional alleles. A large number of MHCII alleles were rare (present in 1-2 individuals), therefore we used for subsequent analyses only alleles that were found in more than three individuals (S2 Fig).” Perhaps these alleles are rare for a reason (susceptibility?).

Page 3, line 69. Please define GLM (and other abbreviations) upon first usage. I was unable to discover where the term was defined in this paper, but I assume it stands for “generalised linear models”.

Page 4, lines 95-96. “Individuals carrying the MHCII DRB allele Migr-DRB*34 are more likely to be AdVpos” is very correctly stated, since it is entirely possible that this class II allele is not causative for susceptibility, rather than a closely-linked gene with an entirely different function.

Page 4, lines 93-97. “We found a negative relationship between microbiome divergence and helminth infection status … indicating that gut microbial communities across helminth infected individuals are more similar than the communities of non-infected individuals.” Assuming that I have understood this correctly, the microbial communities are more similar between individuals infected with helminths compared to those not infected with helminths. So, one interpretation is that different pathogens drive the appearance of different microbial communities, so that “individuals not infected with helminths” could be infected with various other pathogens, each leading to a different microbial community and thus to the greater inter-individual differences in the microbial communities.

Page 4, lines 116-118. “Our results revealed that sequence divergence at MHCIdistPSS … (but not MHCInST) was a significant predictor of distance between samples in the constrained ordination space. ” Perhaps I haven’t understood the meanings of the terms correctly, but from my naïve perspective, different supertypes have different sequences in the peptide binding site, so having more supertypes would lead to more sequence divergence overall (that is, the two measures are correlated). If so, then this sentence points out an unexpected result that needs to be discussed further.

Page 4, lines 120-122. “We observed shifts in microbial composition according to the presence of eight MHCI supertypes (S5 Fig) and four MHCII alleles (S6 Fig).” Is there a reason not to show the microbial compositions of the supertypes and alleles for which the results were (apparently) not significant?

Page 5, lines 133-137. “We identified core taxa (prevalence > 60%) associated with specific MHCI supertypes and MHCII alleles (using analysis of composition of microbes (ANCOM)) and that differed most with respect to their association with high or low estimates of MHC functional diversity (using a differential ranking approach) (see Materials and Methods).” Again my apologies for not understanding the terms used, but what is meant by “prevalence”? If a core taxon needs to be at least 60% of a microbial community, then there can’t be more than one of them per individual.

Page 5, lines 138-140. “In fact, except for one MHCII allele (Migr-DRB*36) which was associated with seven core ASVs (and with 22 ASVs in total) …”. Please define abbreviations upon first use in text. I was unable to find a definition anywhere in the manuscript (and sadly I wasn’t able to guess what it meant).

Page 5, lines 142-144. “This suggests that Migr-DRB*36 is a generalist [39] that indiscriminately binds a high number of antigens affecting both commensal and potentially pathogenic taxa.” If I have understood this interpretation correctly, the number of taxa present in the microbial community of an individual would be considered to be due to direct recognition of antigens from those taxa. An alternative is that recognition of antigens from a taxon leads to exclusion (killing?); if this interpretation is correct, then DRB*36 would be a specialist binding antigens from only those taxa that are not present in such individuals.

Page 5, lines 149-150. “We used Bifidobacterium as a reference frame since this taxa was present in most samples …”. Isn’t “taxon” the singular of “taxa”?

Page 5, lines 151-160. “Our analyses indicate that resistance to AdV was provided by an increase in Campylobacter …”. I didn’t see where the enrichment of Oridobacter in individuals with ST*2 is shown in Fig. 4A, or increase in Olsenella .... associated to MHCInST is shown in Fig. 4C (and so on).

Page 7, legend to Fig. 5. “Associations supported by model selection (AIC > 2) are reported with a red solid line, shaded area represent 95% confidence intervals.” I found a red solid line only in C, so what are dashed black lines in A and B?

Fig. 2. The y-axes of all the graphs in this figure are “infection probabilities”, but the data is shown as one dot for each individual. It is difficult for me to understand how an individual has an “infection probability” since each individual was scored for being infected or not being infected. Please explain.

Figs. 3 and 5 (and S2, S3 and S5). Even with high resolution downloads and with zooming in, it was difficult for me to read some of the labels and designations. At least in the old days, there was a minimum font size for text in figures. Please increase font and choose colours so that the figure conveys the information as intended.

Reviewer #3: Minor remarks

l. 151-160: it is a bit unclear why the authors do not mention all groups whose Cohen’s D 95% CI do not overlap with 0.

l. 402: I assume but it is not stated clearly here that all the 143 anesthetized animals were confirmed to be different individuals by genotyping?

PLOS authors have the option to publish the peer review history of their article (what does this mean?). If published, this will include your full peer review and any attached files.

Reviewer #1: No

Reviewer #2: **Yes: **Professor Jim Kaufman

Reviewer #3: **Yes: **Sebastien Calvignac-Spencer
---

## [Decision Letter · Decision Letter 1]

16 Sep 2021

Dear Dr. Montero,

Thank you very much for submitting your manuscript "Evidence of MHC class I and II influencing viral and helminth infection via the microbiome in a non-human primate" for consideration at PLOS Pathogens. As with all papers reviewed by the journal, your manuscript was reviewed by members of the editorial board and by several independent reviewers. The reviewers appreciated the attention to an important topic. Based on the reviews, we are likely to accept this manuscript for publication, providing that you modify the manuscript according to the review recommendations.

There is an outstanding concern raised by one reviewer. Please add text to the discussion of the manuscript to address this concern related to a weakness of the study and the difficulty in experimentally addressing this concern with wild animals.

Sincerely,

Jason M. Brenchley

Guest Editor

PLOS Pathogens

P'ng Loke

Section Editor

PLOS Pathogens

Kasturi Haldar

Editor-in-Chief

PLOS Pathogens

orcid.org/0000-0001-5065-158X

Michael Malim

Editor-in-Chief

PLOS Pathogens

orcid.org/0000-0002-7699-2064

There is an outstanding concern raised by one reviewer. Please add text to the discussion of the manuscript to address this concern related to a weakness of the study and the difficulty in experimentally addressing this concern with wild animals.

Reviewer Comments (if any, and for reference):

Reviewer's Responses to Questions

**Part I - Summary**

Reviewer #1: The study focuses on an area of tremendous interest; the contribution of host genetic variability, particularly within the major histocompatibility complex, in shaping the composition of the gut microbial community. The premise is particularly important and the authors combine modern laboratory techniques with ﬁeld work, which is both novel and highly interesting.

Reviewer #2: (No Response)

Reviewer #3: The authors have appropriately addressed all my concerns. It is a nice piece that in my view warrants publication in PLoS Pathogens.

**Part II – Major Issues: Key Experiments Required for Acceptance**

Reviewer #1: The authors have not addressed the possible confounding effect of pre-existing immunity (due to prior non-persistent exposure) to AdV or helminths. They simply removed mention of viral persistence from the manuscript to get around my comment. The argument being made in the manuscript is that at this specific moment in time, these specific animals are infected/colonized with these specific “parasites”, because of supposed crosstalk between host MHC, the gut flora, and these “parasites”.

The authors discuss everything (e.g., T-regs, complement, thymic selection, TCR) except the confounding effect of prior exposure on the stratification of the study population into groups.

There must be three or four biologically distinct groups:

A infected; no prior exposure

B. infected; prior exposure

C. uninfected; no prior exposure

D. uninfected; prior exposure

This manuscript only assumes A and C.

To make claims about modified susceptibility to infection and not at all discuss or show data speaking to prior exposure and pre-existing immunity seems disingenuous at worst and a major oversight at best.

At minimum, add an entire paragraph to the discussion that incorporates how prior exposure and pre-existing immunity to AdV and helminths might confound your interpretation.

Reviewer #2: (No Response)

Reviewer #3: NA

**Part III – Minor Issues: Editorial and Data Presentation Modifications**

Reviewer #1: (No Response)

Reviewer #2: Many thanks to the authors for considering my comments and for the modifications which I believe strengthened an already fine paper. I have only two more comments.

Lines 163-164 of the revised manuscript: Either "this taxon was" or "these taxa were", please.

Lines 660-661 of the revised manuscript: many thanks for the kind acknowledgement, but I think that is only one anonymous reviewer.

Reviewer #3: NA

PLOS authors have the option to publish the peer review history of their article (what does this mean?). If published, this will include your full peer review and any attached files.

Reviewer #1: No

Reviewer #2: **Yes: **Prof. Jim Kaufman

Reviewer #3: **Yes: **Sebastien Calvignac-Spencer

Figure Files:

Data Requirements:

Reproducibility:

References:

---

## [Editor Report · Decision Letter 2]

5 Oct 2021

Dear Dr. Montero,

We are pleased to inform you that your manuscript 'Evidence of MHC class I and II influencing viral and helminth infection via the microbiome in a non-human primate' has been provisionally accepted for publication in PLOS Pathogens.

Best regards,

Jason M. Brenchley

Guest Editor

PLOS Pathogens

P'ng Loke

Section Editor

PLOS Pathogens

Kasturi Haldar

Editor-in-Chief

PLOS Pathogens

orcid.org/0000-0001-5065-158X

Michael Malim

Editor-in-Chief

PLOS Pathogens

orcid.org/0000-0002-7699-2064
---

## [Editor Report · Acceptance letter]

3 Nov 2021

Dear Dr. Montero,

We are delighted to inform you that your manuscript, "Evidence of MHC class I and II influencing viral and helminth infection via the microbiome in a non-human primate," has been formally accepted for publication in PLOS Pathogens.

Best regards,

Kasturi Haldar

Editor-in-Chief

PLOS Pathogens

orcid.org/0000-0001-5065-158X

Michael Malim

Editor-in-Chief

PLOS Pathogens

orcid.org/0000-0002-7699-2064